# Constructing high-efficiency orange-red thermally activated delayed fluorescence emitters by three-dimension molecular engineering

Lei Hua[1,2], Yuchao Liu [3], Binbin Liu[2], Zhennan Zhao[1,2], Lei Zhang[2], Shouke Yan [1,3] & Zhongjie Ren [1,2] ✉

Preparing high-efficiency solution-processable orange-red thermally activated delayed fluorescence (TADF) emitters remains challenging. Herein, we design a series of emitters consisting of trinaphtho[3,3,3]propellane (TNP) core derivatized with different TADF units. Benefiting from the unique hexagonal stacking architecture of TNPs, TADF units are thus kept in the cavities between two TNPs, which decrease concentration quenching and annihilation of long-lived triplet excitons. According to the molecular engineering of TADF and host units, the excited states can further be regulated to effectively enhance spin-orbit coupling (SOC) processes. We observe a high-efficiency orange-red emission at 604 nm in one instance with high SOC value of 0.862 cm$^{-1}$ and high photoluminescence quantum yield of 70.9%. Solution-processable organic light-emitting diodes exhibit a maximum external quantum efficiency of 24.74%. This study provides a universal strategy for designing high-performance TADF emitters through molecular packing and excited state regulation.

With a theoretical internal quantum efficiency (IQE) of 100%, thermally activated delayed fluorescence (TADF) materials without noble heavy metals are considered the next-generation emitting materials for organic light-emitting diodes (OLEDs)[1–6]. Characterized by the reverse intersystem crossing (RISC) process from triplet state ($T_1$) to singlet state ($S_1$), TADF emitters can thus achieve high-performance OLEDs. To realize the RISC process, the energy gap between the singlet and triplet states ($\Delta E_{ST}$) of emitters is normally required to be less than 0.2 eV[7]. The small $\Delta E_{ST}$ facilitates the triplet-to-singlet up-conversion process and thus potentially increases the device efficiency[8,9]. Furthermore, due to the continued focus on low-cost production, solution-processable TADF emitters are much more attracting[10]. With the rapid progress of materials chemistry, many kinds of solution-

processable TADF emitters are successfully prepared already, especially for blue-green emitters[11–15]. For instance, the maximum external quantum efficiency (EQE$_{max}$) of up to 27.13% for blue-green solution-processed OLEDs has been obtained[16]. However, the performance of solution-processable orange-red TADF emitters with a wavelength ranging from 560 to 630 nm are significantly unsatisfactory comparing with the blue-green ones. Orange-red emitters tend to transit in a stronger nonradiative pathway due to their excited state energy close to the ground state, resulting in lower device efficiency[17–19]. As an important part of the panchromatic spectrum, high-performance orange-red TADF materials are still far from being well-developed[20,21].

In order to reduce the losses caused by nonradiative transitions to cross the performance barriers of orange-red TADF

[1]State Key Laboratory of Chemical Resource Engineering, Beijing University of Chemical Technology, Beijing 100029, China. [2]Beijing Advanced Innovation Centre for Soft Matter Science and Engineering, Beijing University of Chemical Technology, Beijing 100029, China. [3]Key Laboratory of Rubber-Plastics, Ministry of Education, Qingdao University of Science & Technology, Qingdao 266042, China. ✉e-mail: renzj@mail.buct.edu.cn

emitters, molecular engineering has been used to improve their efficiencies. Among them, synthesizing the rigid TADF molecules with highly conjugated donors and acceptors is a common design strategy[19,22]. In particular, acceptor units with distinct planar features, such as benzothiadiazole[23], dibenzophenazine[24], heptazine[25], dicyanopyrazine[26], phenanthrene[27], anthraquinone[28], and naphthalimide[29] are adopted. As a result, the EQE$_{max}$ values have achieved over 30% for red OLEDs fabricated by vacuum deposition technology[30]. However, the EQE$_{max}$ value of solution-processed small molecule-based orange-red OLEDs is only 23.7%[31], which falls clearly behind the vacuum-deposited devices. Therefore, the development of high-efficiency solution-processed orange-red OLEDs is particularly desirable. In addition, to synthesize the rigid orange-red TADF emitters, regulating their packing modes is another effective method of preparing high-efficiency orange-red light-emitting molecules. Typically, the intermolecular hydrogen bonds are usually designed to control donor-acceptor configurations for reducing nonradiative transition[5,22,32,33]. For example, ref. 34. constructed a stiff three-dimensional supramolecular framework via the intermolecular C-H and C-O hydrogen bonds to suppress molecular vibrations and rotations, which contribute to nonradiative decay. A T-type red emitter featuring a regular arrangement of donor and acceptor through intermolecular stacking was also reported, which considerably increases carrier transport properties and, thus, OLEDs performance[35]. However, these methods of regulating molecular stacking are not universal for all kinds of material systems with TADF characteristics. Therefore, regarding of the regulation of molecular packing, a general molecular design strategy is urgently needed for orange-red TADF emitters.

Trinaphtho[3,3,3]propellane (TNP) with a special three-dimensional structure attracts the materials chemists[36]. It displays a hexagonal packing mode, and thus a cavity architecture forms between two TNP molecules[37], which may spatially separate TADF groups to achieve the purpose of reducing the overlap of

intermolecular triplet spin densities. Moreover, TADF units can facially be covalently linked to TNP skeletons by simple molecular engineering, which is applicable to almost all orange-red TADF emitters. Additionally, the multiple reactive sites of TNP can be modified by different functional units as well, which can further decrease the concentration quenching and annihilation of long-lived triplet excitons. Also, the excited states of the final emitters are possibly regulated in this way. Meanwhile, the intersystem crossing process of the excited states can probably be tuned and thus enhance spin–orbit coupling (SOC) between singlet and triplet states and light-emitting efficiency according to Fermi's golden rule[38–41]. In practice, modulating the energy levels of excited states to enhance the efficiency has been implemented in blue-green emitters[42–45]. There are, however, few reports on the regulation of intersystem crossing processes of orange-red light emitters.

Above all, a design strategy of introducing a specially packed TNP skeleton into the TADF units was proposed to prepare high-efficiency orange-red emitters. Due to the multiple active sites of the TNP skeleton, the type and amount of TADF units or host units can be elaborately controlled. In this way, the excited state energy levels of the emitters can effectively be regulated. Eventually, a series of orange-red TADF emitters were obtained with naphthalimide-dimethylacridine (NAI-DMAc) or tert-butyl modified naphthalimide-dimethylacridine (tBu-NAI-DMAc) as TADF units and 1,3-di-9-carbazolylbenzene (mCP) or dimethylacridine (DMAc) as host units as shown in Fig. 1. The introduction of propeller-like shape TNP can indeed separate TADF units and thus ensure the reduced quenching effect resting on the crystal packing pattern. As a result, through effective molecular-level regulation, tBu-S-mCP presents the best photophysical properties and OLEDs performance. With a small $\Delta E_{ST}$ of 7 meV and a high SOC value of 0.862 cm$^{-1}$, tBu-S-mCP shows an efficient inverse intersystem crossing process with a rate constant of $4.88 \times 10^5$ s$^{-1}$ and a photoluminescence quantum yield (PLQY) of 70.9%. Therefore, solution-processed OLEDs based on tBu-S-mCP exhibit a typical orange-red

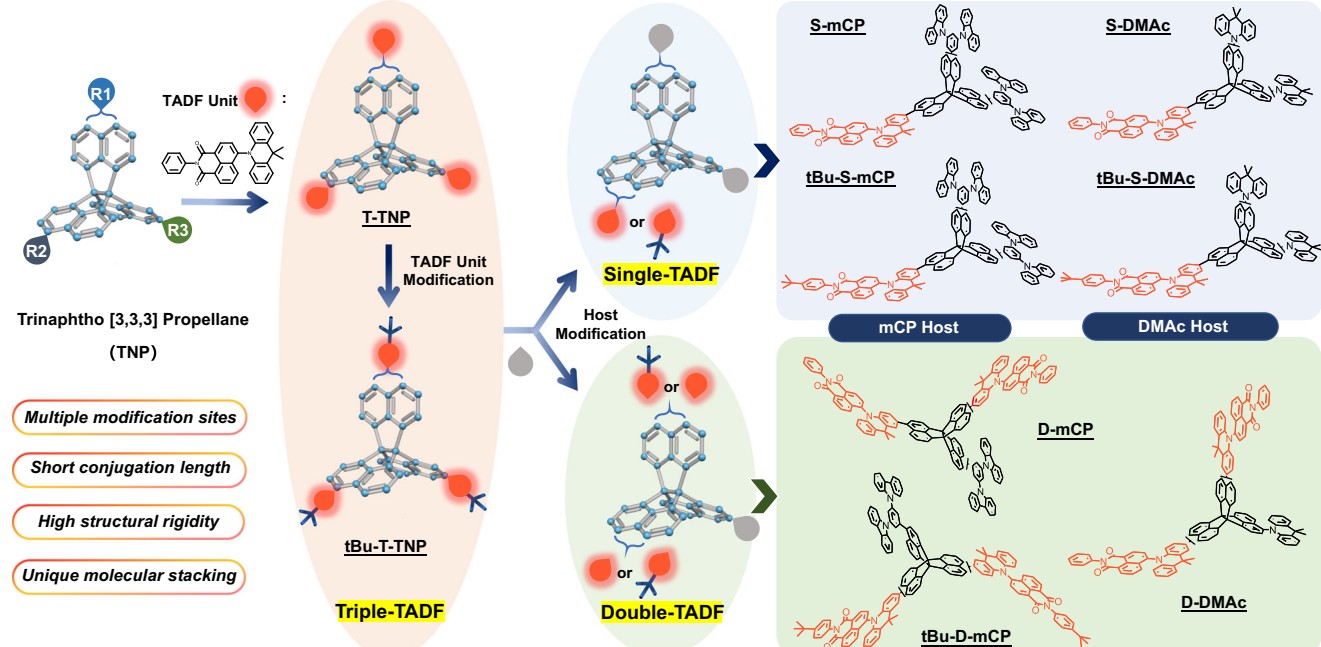

**Fig. 1 | Chemical structure of trinaphtho[3,3,3]propellane (TNP) and molecular engineering pathway.** Three reactive bromo-sites on the TNP molecule enable the introduction of the thermally activated delayed fluorescence (TADF) units (naphthalimide-dimethylacridine (NAI-DMAc) or tert-butyl modified naphthalimide-dimethylacridine (tBu-NAI-DMAc)) or host units (1,3-di-9-carbazolylbenzene (mCP) or dimethylacridine (DMAc)) onto it upon request. Three series of molecules were successfully prepared, named single-TADF series (S-mCP, S-DMAc, tBu-S-mCP, and tBu-S-DMAc), double-TADF series (D-mCP, D-DMAc, and tBu-D-mCP) and triple-TADF series (T-TNP and tBu-T-TNP) according to the number and type of modified TADF and host units.

emission and an impressive EQE of 24.74%, which is the most efficient solution-processed orange-red OLEDs to date. This study provides an idea for the development of high-efficiency orange-red TADF emitters. It reveals that improving molecular packing by introducing multi-dimensional groups and accurately regulating excited state natures through further molecular engineering are promising.

## Results and discussion

### Molecular design and synthesis

As shown in Fig. 1, there are three reactive bromo-sites on the TNP molecule, which enable the introduction of the TADF and host units onto it upon request. The TADF units (NAI-DMAc or tBu-NAI-DMAc) or host units (mCP or DMAc), was modified with a boron ester group, and then attached to the TNP molecule through a Suzuki coupling reaction. It should be noted that since the obtained raw material tribromo-trinaphtho [3,3,3] propellane (TNP-3Br) is a mixture of isomers confirmed by $^1$H nuclear magnetic resonance (NMR) (Supplementary Fig. 2), the position of the bromine substitution cannot be determined and separated, so the series of molecules prepared from this raw material contains structurally closely related isomers. In addition, the effect of tert-butyl groups within the TADF acceptor moiety on the TADF properties was also considered. For single- and double-TADF series, different host units were also introduced to tune the excited state of the obtained TADF emitters. Finally, three series of molecules were successfully prepared, named single-TADF series (S-mCP, S-DMAc, tBu-S-mCP, and tBu-S-DMAc), double-TADF series (D-mCP, D-DMAc, and tBu-D-mCP) and triple-TADF series (T-TNP and tBu-T-TNP) according to the amount and type of the modified TADF and host units as shown in Fig. 1. The details of synthesis steps can be found in Supplementary Fig. 1. The structures of the target molecules were established by $^1$H NMR, $^{13}$C NMR spectroscopies (Supplementary Figs. 3–40) and high-resolution mass spectrometry (Supplementary Figs. 41–51). Furthermore, the purity of all emitters was characterized by high-performance liquid chromatography, which was above 99.9% (Supplementary Fig. 52). All molecules obtained possess good solubility in common organic solvents. In addition, all emitters show no glass transition, melting, or crystallization behavior in the range from room temperature to 300 °C (Supplementary Fig. 53). And their initial thermal decomposition temperatures are all above 400 °C, which indicates that all emitters have good thermal stability and meet the requirement for the device preparation process.

### Frontier molecular orbital simulation

The distributions of the highest occupied molecular orbital (HOMO) and lowest unoccupied molecular orbital (LUMO) of these molecules were calculated by density functional theory (DFT) simulation with the B3LYP 6-31 G(d) level to predict their TADF features as shown in Fig. 2 and Supplementary Fig. 54. The LUMO orbitals are totally located on the part of electron acceptor naphthalimide in NAI-DMAc units. The energy levels of LUMO for the nine molecules are in the range of −2.57 to −2.61 eV, with slight differences due to the influence of modifying groups.

As the number of TADF units increases, the LUMO energy level tends to decrease. The modification with the terminal tert-butyl group also shows a certain effect on LUMO energy levels, which is shallower than that without tert-butyl modification. For example, the LUMO energy level of tBu-S-mCP is 0.02 eV higher than that of S-mCP. The incorporation of host units, regardless of the type, hardly affects the energy level of LUMO. On the contrary, the modification of host units displays a significant effect on the distribution of HOMO orbitals, while a tiny effect of tert-butyl group within TADF units on HOMO energy levels is identified. For DMAc-modified molecules, such as tBu-S-DMAc, their HOMO orbitals locate on the additionally modified DMAc host unit, so the HOMO energy levels are around −4.88 eV. Except for these three molecules with DMAc host, the HOMO orbitals of the other molecules all distribute on the part of the acridine group within TADF units. The HOMO orbital of the single-TADF molecules extends to one of the three symmetrical parts of the TNP skeleton, which confirms that the non-conjugated TNP core has a tiny effect on the orbital distribution of TADF units. The HOMO orbitals of both tBu-S-mCP and S-mCP are around −5.13 eV. The HOMO orbits of the double-TADF series extend to two groups of acridine units within TADF through a non-conjugated TNP core, meanwhile, the triple-TADF series extend to all acridine in TADF units and TNP. Therefore, with the expansion of the HOMO orbitals, from single-TADF units to three TADF units, the HOMO energy level is significantly elevated from −5.13 to −5.00 eV. In addition to the DMAc-modified molecules, the energy gap ($E_g$) also tends to decrease from the single-TADF series to the triple-TADF series affected by the HOMO energy level, which indicates the more red-shifted emission for the triple-TADF series. Due to the elevated LUMO energy level, the $E_g$ of the molecules without tert-butyl modification become narrower, and thus the emission will theoretically be red-shifted. In addition, the HOMO and LUMO distributions of all

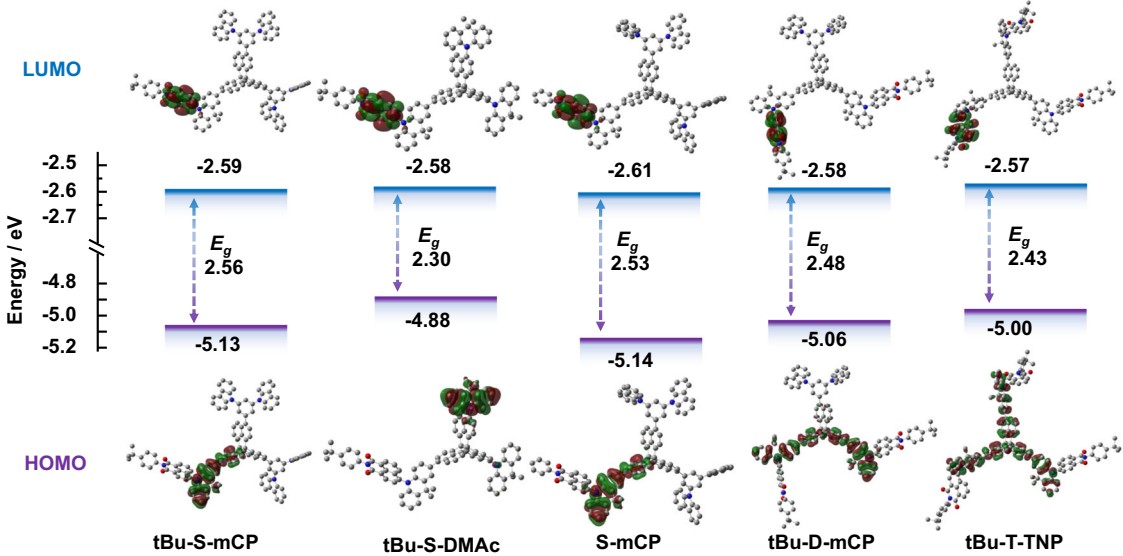

**Fig. 2 | Frontier molecular orbital simulation.** The distributions of the highest occupied molecular orbital (HOMO) and lowest unoccupied molecular orbital (LUMO) with their energy gaps ($E_g$s) of five typical molecules were calculated by density functional theory (DFT) simulation with the B3LYP 6-31 G(d) level.

molecules display the obvious separation with a small overlapping integral, which ensures a very small energy splitting between the singlet state ($S_1$) and the triplet state ($T_1$), guaranteeing the high-efficiency intersystem crossing process.

## Molecular packing

A single crystal of T-TNP was obtained by slow solvent evaporation. The single crystal structure and molecular packing pattern of T-TNP are shown in Fig. 3 and Supplementary Fig. 55. The introduction of propelleranes endows the distinctive molecular packing of T-TNP. That is, the three-dimensional core ensures the two adjacent blades in a triangular shape perfectly. At the same time, the peripheral TADF units are exactly located in the cavity formed between the propelleranes blades. No directly close face-to-face packing from TADF units is observed, which can effectively suppress the intermolecular electron-exchange interactions. The distance between naphthalimide group in TADF units and naphthalene rings in the TNP blade is 3.368 Å, indicating weak π-π stacking of adjacent TADF units, which is beneficial to reduce the concentration quenching effect caused by the high concentration of local triplet excitons. In addition, the C-H...π interactions between phenyl rings and acridine units of adjacent TADF units with a 2.788 Å distance can be obtained, which probably helps to lock molecular architecture and thus reduces nonradiative deactivation of excitation. Furthermore, benefiting from the large intermolecular cavity, this packing structure hardly affects the torsion angle between the donor and acceptor of TADF units. The twist angle between naphthalimide and acridine groups on the three blades maintains to be 85.15°, which is consistent with the strong separation between HOMO and LUMO in the frontier molecular orbital simulation.

Intermolecular interactions can be observed from the crystal packing structure in Fig. 3b, and the layered packing architecture can be clearly seen. The formation of multiple C-H...π interactions between the neighboring emitters with distances in the range of ca. 2.824–3.536 Å is prone to fix the molecular motion and decreases the nonradiative transition. Moreover, in this stacking mode, the intermolecular interactions between two TADF units in two layers can also be effectively weakened to suppress exciton quenching.

## Photophysical properties

The ultraviolet–visible (UV-Vis) absorption and fluorescence emission spectra of nine molecules were measured in dilute toluene solution and in nondoped films. As shown in Fig. 4a, in toluene solution, single-TADF, double-TADF, and triple-TADF series show similar UV-Vis absorption and fluorescence emission spectra. All emitters exhibit double-peak absorptions at 280–400 nm, which can be attributed to the π-π transitions. Since the absorption peaks of carbazole and acridine mainly appear below 300 nm[29,46], with the increased content of host units, the absorption peak at 290 nm becomes stronger for the single-TADF series, while the absorption peak at 350 nm is higher for the triple-TADF series. Moreover, the modification of tert-butyl groups nearly doesn't affect absorption spectra in dilute solutions. Similarly, the fluorescence emission spectra in solution are also mainly affected by the amount of TADF units. All the molecules show orange-red emission at 590−600 nm. The emission peak of single-TADF series, such as tBu-S-mCP, is located at 591 nm, while the emission peak of triple-TADF series, such as tBu-T-TNP, red-shifts to 597 nm. This is consistent with the narrowing of $E_g$ owing to the expansion of HOMO orbitals from the frontier molecular orbital (FMO) simulation. Additionally, a weak emission at 425 nm can be observed from the host units, especially for the single-TADF series. The UV-Vis absorptions and fluorescence emissions in pure films are basically similar to those in the toluene solution. However, the effect of tert-butyl groups in TADF units on the intermolecular interaction in the aggregated state can obviously be found. In the same series, both the red-shifted absorption edge and emission of the emitters without tert-butyl modification are significantly observed by comparing with the tert-butyl modification counterpart, especially for the triple-TADF series. The fluorescence emission peak of T-TNP films is red-shifted by 9 nm compared to tBu-T-TNP. Furthermore, an absorption tail band over 400 nm can be detected in the film's state, which can be attributed to the absorption of the intramolecular charge transfer state. As shown in Fig. 4c and Supplementary Fig. 56, the phosphorescence spectra also display the unfeatured vibrational structure similar to the fluorescence spectra, implying the charge transfer characteristic of $T_1$. The $\Delta E_{ST}$s of all emitters were measured by fluorescence and phosphorescence spectra in toluene solution at 77 K. The onset positions of the fluorescence and phosphorescence spectra at 77 K were used as the singlet and triplet energy levels, respectively. Similar to the FMO simulations results, the $\Delta E_{ST}$s of all the emitters are less than 30 meV, which ensures an efficient intersystem crossing process.

The PLQYs of all emitters were measured for the doped films, as shown in Fig. 4d. PLQYs in vacuum were converted by PLQYs in air and the integral ratio of the steady-state emissions of the corresponding films in vacuum to air (Supplementary Fig. 57). The regulation results of molecular engineering are fully demonstrated in the performance of PLQY. The emitters containing tert-butyl group present a significantly higher PLQY, which may be attributed to suppressing intramolecular vibrational relaxation by tert-butyl group and reducing aggregation-caused quenching[47]. And with the increased ratio of TADF units, the PLQY shows a downward trend; in the same series, the PLQYs of mCP-modified emitters are also higher than that of DMAc-modified ones. In addition, PLQYs of the emitters doping with the different ratios host have been characterized. Taking tBu-S-mCP as an example, 2, 5, and 10 wt% tBu-S-mCP are doped with 5 wt% poly(N-vinylcarbazole) (PVK)

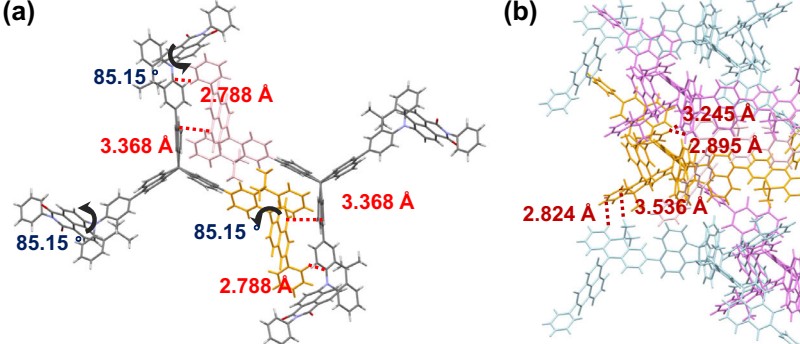

**Fig. 3 | Single crystal structure of T-TNP. a** The bimolecular structure. The twist angle between naphthalimide and acridine groups on the three blades maintains to be 85.15°. Two adjacent thermally activated delayed fluorescence (TADF) units show certain interactions only at the edges. The interactional distances within 4.0 Å are marked. **b** The layered packing architecture of T-TNP and the interactions between layers. The interactional distances within 4.0 Å are marked.

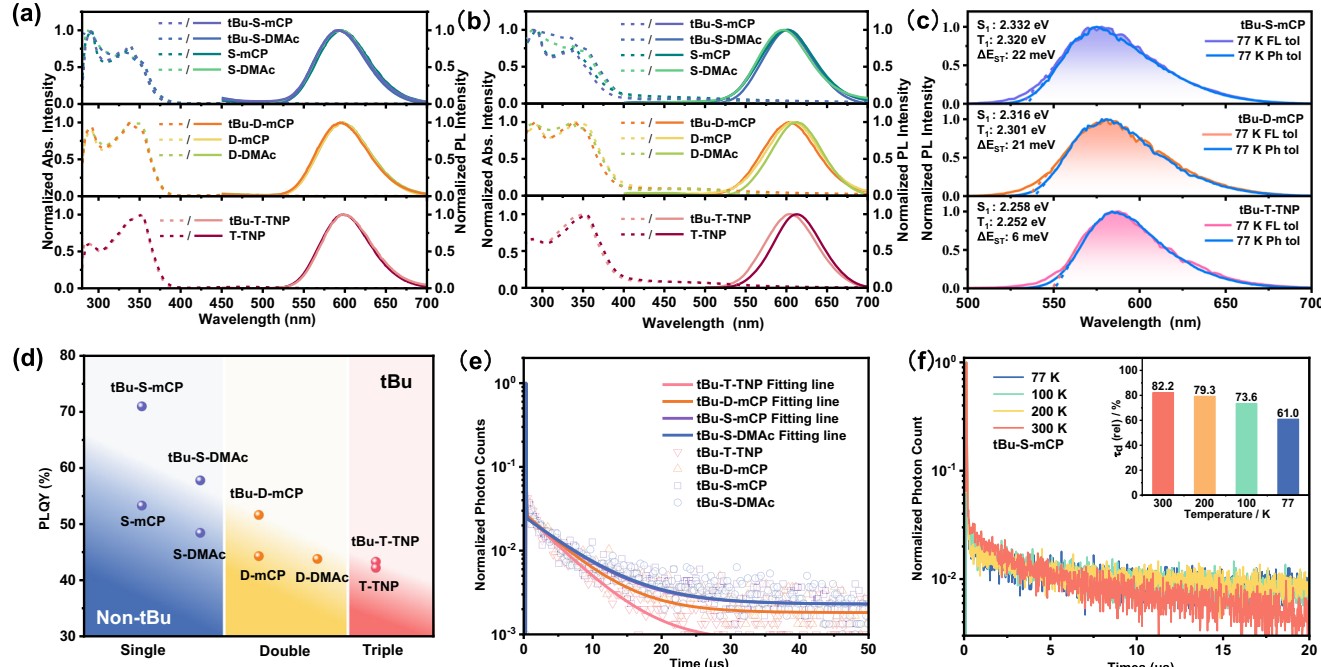

**Fig. 4 | Photophysical properties.** Ultraviolet–visible absorption (Abs.) spectra (dotted line) and photoluminescence (PL) spectra (solid line) of emitters **a** in toluene solution and **b** in pure films. **c** Fluorescence (FL) and phosphorescence (Ph) spectra of three typical emitters detected in toluene at 77 K. Singlet energy level ($S_1$), triplet energy level ($T_1$), and their energy difference ($\Delta E_{ST}$) are marked. **d** Photoluminescence quantum yield (PLQY) in a vacuum of 2 wt% emitter doped in 5 wt% Poly(*N*-vinylcarbazole) (PVK) and 93 wt% 9H-carbazole-3-carbonitrile (mCP-CN) (converted by the PLQYs of blended films in air and the integral ratio of the steady-state emissions of the corresponding films in vacuum to air). Obviously, tert-butyl modified emitters (tBu) locate on the top and non-tert-butyl modified emitters (Non-tBu) distribute on the bottom. **e** Transient PL spectra of the blended emitter films in a vacuum (the dotted lines are the measured results, and the solid lines are the fitting results). **f** Temperature-dependent transient PL decay spectra of the blended tBu-S-mCP films (The inset shows the proportion of delayed fractions ($\tau_d$) at different temperatures).

and the corresponding ratio 9H-carbazole-3-carbonitrile (mCP-CN). It is found PLQY of tBu-S-mCP in vacuum decreases by about 7% (from 70.9 to 63.1%) when the doping concentration improves from 2 to 10 wt% (Supplementary Fig. 58). However, PLQY of the control emitter (NAI) reduces by ca. 14% with a doping concentration of from 1.5 to 6 wt%[29]. PLQY of tBu-S-mCP is significantly less affected by the doping concentration. In combination with the characterization of the single crystal, it is well demonstrated that the architecture of TNP can effectively suppress exciton quenching.

From the fluorescence lifetime curves, all the emitters of the doped films show double exponential decays, including prompt fluorescence (PF) of more than 20 ns and delayed fluorescence (DF) components of several microseconds (Fig. 4e, Supplementary Fig. 59 and Table 1). Judging from the amount of TADF units, the fewer TADF units present, the longer fluorescence lifetime and the higher delayed fluorescence ratio. The fluorescence lifetime of the single-TADF series is over 6 μs with a delayed components ratio of above 81%, which is slightly decreased for the double-TADF series and reduces to only ca. 5 μs for the triple-TADF series. This result possibly accounts for the strong concentration quenching effect in the triple-TADF series, resulting in the larger nonradiative transition rates and the shorter triplet exciton lifetimes. And thus, T-TNP displays the lowest PLQY among of nine emitters. With the decreased content of TADF units, the nonradiative transition rate also shows a gradual reduction. In terms of RISC rate ($k_{RISC}$), the modification of tert-butyl group also influences it obviously. The $k_{RISC}$s of all tert-butyl-modified emitters are significantly higher than the non-modified counterpart. With similar structures, $k_{RISC}$ of tBu-S-mCP is 1.3 times faster than S-mCP, and tBu-D-mCP is 1.2 times faster than D-mCP. Furthermore, the type of host doesn't change $k_{RISC}$ significantly. In summary, among the single-TADF series, tBu-S-mCP containing tert-

butyl group and mCP host demonstrates the highest PLQY of 70.9% and the fastest $k_{RISC}$ of $4.88 \times 10^5 \, s^{-1}$.

Fluorescence decay of the doped films was determined at different temperatures to characterize the TADF feature. The DF components of all emitters exhibit the apparent temperature dependence, and the proportion of DF components increases significantly as the temperature is elevated from 77 to 300 K, indicating the typical TADF properties of all emitters (Supplementary Fig. 60). Typically, the DF ratio of tBu-S-mCP increases from 61% at 77 K to 82.2% at 300 K as shown in Fig. 4f, which indicates the much more triplet exciton of tBu-S-mCP contributes to the luminescence at 300 K than that of 77 K. That is, tBu-S-mCP presents a more efficient RISC process at 300 K than 77 K, which is consistent with the nature of thermal activation.

### Natural transition orbits calculation

It is obvious that tBu-S-mCP manifests excellent photophysical properties among the nine emitters. Although all of the emitters own a similar propellerane core, the different photophysical properties are correlated to the molecular engineering of the periphery. In order to accurately figure out the regulatory effect of multiple molecular engineering from the perspective of excited states, the excited states of each emitters were calculated using time-dependent density functional theory (TD-DFT) and Multiwfn[48] were utilized to analyze the natural transition orbits (NTOs). Furthermore, spin–orbit coupling matrix element (SOCME) calculations between singlet and triplet states were also carried out using ORCA 4.1.1 package[49] at B3LYP/G TZVP.

Figure 5 shows the NTOs corresponding to the lowest singlet states ($S_1$) and triplet state ($T_n$, $n$ = 1, 2, 3, 4…) on which the maximum SOC of the five emitters is located. The complete excited state energy levels and SOC constants between excited states can be found in

**Table 1 | Summary of photophysical properties of the nine emitters**

| | PLQY[a] | $\tau_p$/ns[b] | $\tau_d$/µs[c] | $\varphi_p$[d] | $\varphi_d$[e] | $k_P$/$10^6$ s$^{-1}$[f] | $k_{DF}$/$10^5$ s$^{-1}$[g] | $k_{ISC}$/$10^6$ s$^{-1}$[h] | $k_{RISC}$/$10^5$ s$^{-1}$[i] | $k^s_{nr}$/$10^5$ s$^{-1}$[j] |
|---|---|---|---|---|---|---|---|---|---|---|
| tBu-S-mCP | 70.9% | 23.9 | 6.41 | 18.5% | 81.5% | 5.48 | 9.02 | 4.47 | 4.88 | 2.95 |
| tBu-S-DMAc | 57.8% | 23.4 | 6.65 | 17.1% | 82.9% | 4.22 | 7.20 | 3.50 | 4.21 | 3.06 |
| S-mCP | 56.3% | 23.4 | 6.47 | 19.1% | 80.9% | 4.59 | 7.05 | 3.72 | 3.70 | 3.82 |
| S-DMAc | 49.5% | 24.7 | 6.49 | 17.3% | 82.7% | 3.47 | 6.30 | 2.87 | 3.64 | 3.04 |
| tBu-D-mCP | 51.7% | 23.3 | 5.59 | 19.2% | 80.8% | 4.26 | 7.47 | 3.45 | 3.89 | 3.96 |
| D-mCP | 44.4% | 25.6 | 5.79 | 19.4% | 80.6% | 3.35 | 6.18 | 2.70 | 3.19 | 3.61 |
| D-DMAc | 43.9% | 26.3 | 5.57 | 19.5% | 80.5% | 3.24 | 6.35 | 2.61 | 3.26 | 3.54 |
| tBu-T-TNP | 43.4% | 25.9 | 5.33 | 20.1% | 79.9% | 3.36 | 6.51 | 2.69 | 3.24 | 3.82 |
| T-TNP | 42.3% | 23.4 | 4.69 | 23.6% | 76.4% | 4.27 | 6.89 | 3.26 | 2.92 | 5.82 |

[a]The photoluminescence quantum yields (PLQYs) of blended films in vacuum (converted by the PLQYs of blended films in air and the integral ratio of the steady-state emissions of the corresponding films in vacuum to air).
[b]The lifetime of prompt fluorescence component in vacuum.
[c]The lifetime of delayed fluorescence component in vacuum.
[d, e]The ratio of (d) prompt fluorescence (PF) component and (e) delayed fluorescence component determined by fitting transition decay curves in vacuum.
[f,g]The (f) prompt and (g) delayed fluorescence decay rate, respectively.
[h]The rate constants of intersystem crossing.
[i]The rate constants of reverse intersystem crossing.
[j]The nonradiative rate constants of the lowest singlet state ($S_1$).

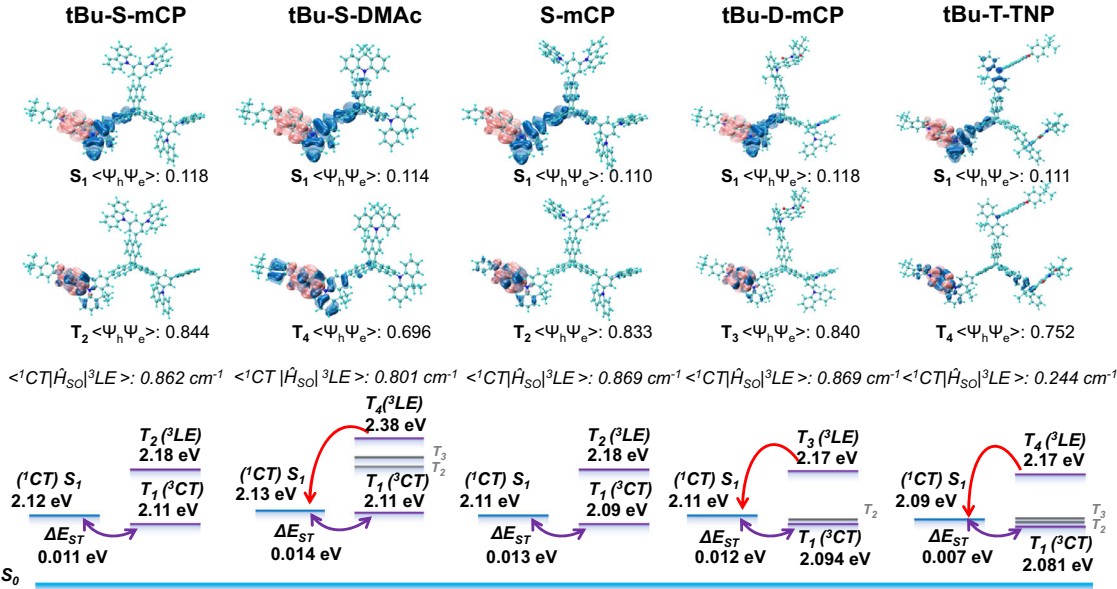

**Fig. 5 | NTO (natural transition orbits) calculation.** Energy level diagram of excited states of five typical emitters ($S_0$ is the ground state; $S_1$ is the lowest singlet excited state and $T_n$ ($n$ = 0, 1, 2, 3, 4…..) is corresponding triplet excited state; $^1CT$ is charge transfer singlet state; $^3LE$ is locally excited triplet state; $\Delta E_{ST}$ is the energy gap between the lowest singlet and triplet states), their NTO analysis (blue areas are holes and pink areas are electrons; $<\psi_h\psi_e>$ is the overlap integral of hole and electron orbital) and spin–orbit coupling matrix elements of $^1CT$ and $^3LE$ ($<^1CT|\hat{H}_{SO}|^3LE>$).

Supplementary Fig. 61 and Supplementary Table 1. Except for D-DMAc and T-TNP, all triplet excited states that dominate the spin–orbit coupling process are all from the localized excited states ($^3LE$), which conform to the El-Sayed rule[50–52]. Due to the predominance of $^3CT$ states, the corresponding SOC values of D-DMAc and T-TNP are quite small (<0.1 cm$^{-1}$) and almost negligible, which accounts for their inefficient RISC process and the lowest PLQY among the nine emitters.

Regarding controlling the amount of TADF units, much more TADF units would explicitly bring the energy level degeneracy of the lowest excited state. For instance, the difference between $T_1$ and $T_2$ of tBu-D-mCP is only 0.01 eV, and $T_1$-$T_3$ of tBu-T-TNP are all degenerate energy levels. The spin-flip process between degenerate energy levels is inherently inefficient, and the appearance of degenerate energy levels blocks the channel for the flip process from triplet-to-singlet states. In terms of SOCME, the SOC value of the double-TADF and

triple-TADF series appears 0 cm$^{-1}$ in the lower energy levels. Compared with the single-TADF series, although the dominant spin–orbit flip process is similar, the reduced channels and the intramolecular energy loss due to the increased degenerate energy levels, which attributes to the inferior photophysical properties of double-TADF and triple-TADF series. The single-TADF series were analyzed individually to demonstrate the regulatory role of tert-butyl and host modifications. As shown in Fig. 5, these four emitters are all dominated from $^3LE$ to $^1CT$ state in the spin-flip process, and the SOC constants are all exceeding 0.8 cm$^{-1}$, which can establish an effective intersystem crossing channel. Moreover, the type of modified host units also affects the corresponding excited state energy level of $^3LE$. The excited state energy level of the maximum SOC constant corresponding to tBu-S-DMAc and S-DMAc falls on $T_4$, while the $^3LE$ state corresponding to mCP-modified tBu-S-mCP and S-mCP locates on $T_2$. Higher-level transitions may lead

to more serious internal conversion energy loss. The energy level difference between $^3$LE and $^1$CT of tBu-S-mCP is only 0.06 eV, while the corresponding energy level difference of tBu-S-DMAc is 0.25 eV. According to the second-order vibronic coupling mechanism, the $^3$LE state can act as a mediator role who can assist in the flipping of the exciton from the triplet state to the singlet state[53]. The higher energy level difference brings the lower RISC efficiency[54]. Consequently, although they have the same high SOC values and narrow $\Delta E_{ST}$, the RISC process of tBu-S-mCP is more efficient than tBu-S-DMAc. This result manifests the regulation effect of mCP host on $^3$LE state in the comparison of S-mCP and S-DMAc. Additionally, tert-butyl group hardly affects the NTOs of triplet and thus tBu-S-mCP and S-mCP display the almost same $^3$LE. However, the regulation effect of tert-butyl group is mainly reflected in the singlet state. Compared with S-mCP, the $^1$CT state energy level of tBu-S-mCP is slightly higher. The slightly high $^1$CT not only makes the emission of tBu-S-mCP blue-shift, but also brings a smaller energy level difference between $^3$LE and $^1$CT, which slightly promotes the intersystem crossing process as well. Therefore, controlling the amounts of TADF units can effectively regulate the SOC channel, while the host engineering facilely acts on the regulation of $^3$LE, and the tert-butyl modification can tune the $^1$CT. The multiple regulations successfully endow the $^3$LE energy level of tBu-S-mCP only 0.06 eV higher than the $^1$CT energy level, with multiple SOC channels and a high SOC value 0.862 cm$^{-1}$ of <$^1$CT$|\hat{H}_{SO}|^3$LE>. All these properties ensure an efficient intersystem crossing process of tBu-S-mCP and, thus excellent theoretical performance, which also demonstrates the validity of our molecular design strategy.

## Device characteristics

The film-forming ability of the nine emitters was characterized by atomic force microscopy before preparing the solution-processed OLEDs. As seen from Supplementary Fig. 62, all of the emitters show good film-forming ability in the doped host, such as, the doped film of

tBu-S-mCP with $5 \times 5\ \mu m^2$ area exhibits a low surface roughness with Ra of 0.196 nm. Figure 6a shows the solution-processed OLEDs with a configuration of indium tin oxide (ITO)/poly(3,4-ethylenedioxythiophene):poly(styrenesulfonate) (PSS: PEDOT) (30 nm)/the doped emitters (40 nm)/bis[2-(diphenylphosphino) phenyl] ether oxide (DPEPO) (2 nm)/1,3,5-tri[(3-pyridyl)-phen-3-yl] benzene (TmPyPB) (60 nm)/ lithium fluoride (LiF) (0.9 nm)/aluminum (Al) (130 nm), in which PSS: PEDOT served as hole-injecting and hole-transporting layer, LiF acted as an electron-injecting layer, DPEPO used as a hole-blocking layer, respectively, TmPyPB utilized as an electron-transporting layer. For the emitting layer, the emitters were embedded in the hosts of mCP-CN and PVK, which guarantees good exciton dispersion ability and film-forming ability. Furthermore, mCP-CN with a sufficiently high triplet energy level can effectively balance hole-electron injection and transport properties[55]. The energy levels of emitters were determined by cyclic voltammetry measurements, as shown in Supplementary Fig. 63, to make sure good matching with the energy levels of transfer layers.

Similar to the photoluminescence (PL) spectra, the electroluminescence (EL) spectra of all devices range from 594 to 608 nm, typical of orange-red light-emitting devices. As shown in Fig. 6b and Table 2, with the increase in the number of TADF units, the EL spectra of the three-TADF series redshift obviously compared with the single-TADF series, and the EL peaks of both tBu-T-TNP and T-TNP are inclined to be red light at 608 nm. The EL peak of the double-TADF series is located at ca. 604 nm. In the single-TADF series, S-DMAc shows the reddest emission located at 598 nm, and the EL spectra of tBu-S-mCP and S-mCP are blue-shifted, located at 594 nm. The turn-on voltages of all devices are similar, ranging from 5.2 to 5.4 V, as shown in Fig. 6c. At around 7 V, all devices achieve a stable maximum brightness of over 1000 cd m$^{-2}$.

In terms of EQE$_{max}$, as shown in Fig. 6e and Table 2, due to the local exciton quenching of the multi-TADF units and the excessive

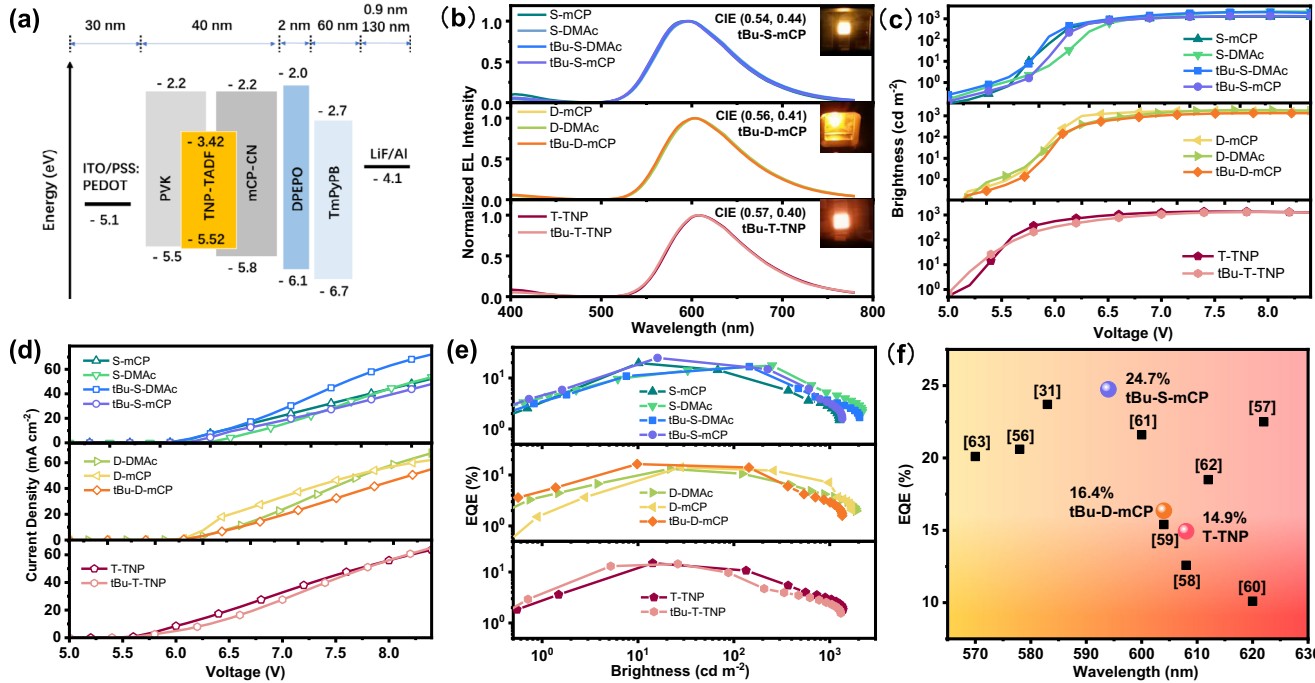

**Fig. 6 | Device characteristics. a** Solution-processed device architecture and related energy levels. ITO indium tin oxide, PSS: PEDOT poly(3,4-ethylenedioxythiophene):poly(styrenesulfonate), TNP-TADF single-TADF series (S-mCP, S-DMAc, tBu-S-mCP, and tBu-S-DMAc), double-TADF series (D-mCP, D-DMAc, and tBu-D-mCP), and triple-TADF series (T-TNP and tBu-T-TNP) emitters; PVK Poly(*N*-vinylcarbazole), mCP-CN 9H-carbazole-3-carbonitrile, DPEPO bis[2-

(diphenylphosphino) phenyl] ether oxide, TmPyPB 1,3,5-tri[(3-pyridyl)-phen-3-yl] benzene, LiF lithium fluoride, Al aluminum. **b** Electroluminescence (EL) spectra detected at maximum brightness. **c** Curves of brightness versus voltage. **d** Curves of current efficiency versus voltage. **e** Curves of external quantum efficiency (EQE) value versus brightness. **f** The reported EQE values of solution-processed devices employing orange-red thermally activated delayed fluorescence emitters.

**Table 2 | Electroluminescence (EL) properties of solution-processed organic light-emitting diodes (OLEDs) based on orange-red emitters**

| | EL$_{peak}$ (nm)[a] | V$_{on}$ (V)[b] | L$_{max}$ (cd m$^{-2}$)[c] | CE$_{max,100,\ 1000}$ (cd A$^{-1}$)[d] | PE$_{max,100,\ 1000}$ (lm W$^{-1}$)[e] | EQE$_{max,100,\ 1000}$%[f] | CIE (x,y)[g] |
|---|---|---|---|---|---|---|---|
| tBu-S-mCP | 594 | 5.4 | 1348 | 59.3, 34.2, 6.57 | 33.3, 18.5, 3.13 | 24.74%, 14.89%, 3.02% | 0.54,0.44 |
| tBu-S-DMAc | 596 | 5.2 | 2047 | 38.5, 38.5, 6.93 | 21.6, 21.6, 3.40 | 16.69%, 16.69%, 3.22% | 0.54,0.43 |
| S-mCP | 594 | 5.4 | 1266 | 45.3, 22.2, 5.36 | 26.3, 12.1, 2.55 | 19.65%, 9.67%, 2.53% | 0.52,0.43 |
| S-DMAc | 598 | 5.2 | 2175 | 39.6, 39.6, 9.62 | 20.7, 20.7, 4.58 | 17.61%, 17.61%, 4.51% | 0.54,0.43 |
| tBu-D-mCP | 604 | 5.4 | 1349 | 33.5, 27.8, 5.40 | 18.8, 15.0, 2.49 | 16.35%, 14.01%, 2.90% | 0.56,0.41 |
| D-mCP | 604 | 5.4 | 1710 | 28.0, 23.4, 6.79 | 15.7, 12.7, 3.44 | 14.17%, 12.17%, 3.68% | 0.56,0.41 |
| D-DMAc | 604 | 5.2 | 1881 | 26.2, 20.4, 6.71 | 14.7, 11.1, 3.20 | 13.22%, 10.56%, 3.69% | 0.56,0.41 |
| tBu-T-TNP | 608 | 5.2 | 1305 | 25.9, 8.21, 3.82 | 15.1, 4.45, 1.71 | 14.42%, 4.71%, 2.31% | 0.57,0.40 |
| T-TNP | 608 | 5.2 | 1376 | 26.3, 18.4, 4.68 | 15.3, 10.3, 2.23 | 14.93%, 10.7%, 2.89% | 0.57,0.39 |

[a]The peak value of electroluminescence at maximum brightness.
[b]Turn-on voltage at 1 cd m$^{-2}$.
[c]Maximum luminance.
[d]Maximum current efficiency and values at 100 and 1000 cd m$^{-2}$.
[e]Maximum power efficiency and values at 100 and 1000 cd m$^{-2}$.
[f]Maximum external quantum efficiency value and values at 100 and 1000 cd m$^{-2}$.
[g]Coordinates of Commission Internationale de L'Eclairage.

intramolecular nonradiative transition losses caused by degeneration of excited state energy levels, the EQE$_{max}$s of devices based on the double-TADF and triple-TADF series are 16.35 and 14.93%, respectively, which are lower than all single-TADF series devices. In the single-TADF series, the device performance of the two emitters with mCP host is better than that with the DMAc host, which confirms that the regulation of $^3$LE state by mCP host is effective, consistent with the simulation of the molecular excited state and photophysical results. The device EQE$_{max}$ of S-mCP without tert-butyl group modification is slightly low (19.65%), consistent with the certain effect of tert-butyl group on the molecular excited state. Similar to most orange-red TADF materials, the emitter does not perform well at high luminance of 1000 cd m$^{-2}$, which is caused by a strong nonradiative transition at higher voltages. Unsurprisingly, the device fabricated from tBu-S-mCP achieves the highest EQE$_{max}$ of 24.74% and the highest power efficiency of 33.3 lm W$^{-1}$ in the series. Meanwhile, the emitting dipole orientations of the typical emitters, tBu-S-mCP and tBu-S-DMAc, were determined by angle- and polarization-resolved PL spectroscopy measurements (Supplementary Fig. 64). Both horizontal dipole ratios (Θ//) are fitted to be 73%, which is nearly consistent with the NAI-DMAc[29]. A higher molecular dipole orientation value is beneficial for enhancing the EQE of OLEDs[4]. According to the horizontal dipole orientation ratios and PLQY, the theoretically predicted EQE$_{max}$ values is 25.02% for tBu-S-mCP by using optical simulation (Supplementary Fig. 65), which is reasonably higher than our experimental data. As shown in Fig. 6f and Supplementary Table 2, tBu-S-mCP-based OLEDs achieve high efficiency among the solution-processed small molecule-based orange-red light-emitting device[31,56–63], which expresses a 10% improvement over previous solution-processed NAI-DMAc devices[57]. In addition, the displayed device results are the best one, all emitters are fabricated at least twice under the same conditions, and the deviation is less than 3% in the test of eight or more pixels during the device fabrication processes. The result fully demonstrates that introducing special three-dimensional structures to improve molecular packing and thus precisely regulating molecular excited state energy levels is an effective design strategy for red-emitting OLEDs.

In summary, the effect of a three-dimensional structural skeleton on the photophysical properties and OLEDs performance of TADF emitters was proposed herein. The orange-red TADF emitters with three-dimensional architectures prepared by attaching TADF units onto TNP units were successfully obtained. The special honeycomb-like packing significantly reduces the overlap of intermolecular triplet spin densities and, thus, obviously, decreases aggregation quenching.

Furthermore, the excited state natures of the obtained emitters can also be tuned by changing TADF and hosts units, sophisticatedly. Finally, the prepared emitter, tBu-S-mCP, displays a high PLQY of 70.9%, a fast $k_{RISC}$ of $4.88 \times 10^5$ s$^{-1}$ and a high SOC value of 0.862 cm$^{-1}$. Therefore, the solution-processed tBu-S-mCP-based OLEDs present an impressive EQE$_{max}$ of 24.74%, which is higher than that of NAI-DMAc-based OLEDs (22.5%). The results suggest that introducing special three-dimensional structures into TADF emitters to precisely control molecular packing provides a universal way for designing high-performance OLEDs materials.

## Methods
### Materials
All materials were purchased from commercial suppliers, such as Bide Pharma tech, Energy Chemical, and were used without further purification, unless otherwise noted. Toluene and tetrahydrofuran were purchased from Beijing Tongguang, and the moisture was removed with sodium reflux for 24 h. Anhydrous 1,4-Dioxane were purchased from J&K Scientific. The host material 9H-carbazole-3-carbonitrile (mCP-CN) used for the doped films was purchased from Lumtec. And poly (n-vinylcarbazole) (PVK) (M$_w$: 90000) was from J&K Scientific. Poly(3,4-ethylenedioxythiophene): poly(styrenesulfonate) (PSS: PEDOT) (4083), 1,3,5-tri[(3-pyridyl)-phen-3-yl] benzene (TmPyPB), lithium fluoride (LiF), and (bis[2-(diphenylphosphino) phenyl] ether oxide) (DPEPO) used in device fabrication were purchased from Xi'an Polymer Light Technology Corp.

### Device fabrication and performance measurement
Commercially purchased ITO glass was ultrasonically cleaned with isopropanol, followed by deionized water, and dried in an oven at 120 °C. After hydrophilic treatment by UV ozone for 6 min, the PSS: PEDOT aqueous solution was spin-coated on the ITO substrates at 3000 rpm. Then, the substrates were transferred into the glovebox and heat-treated at 120 °C to remove the residual water. Subsequently, the mixed emissive layer solution (0.2 mg light-emitting material, 0.5 mg PVK, and 9.3 mg mCP-CN were dissolved in 1 mL anhydrous chlorobenzene) was spin-coated on the PSS: PEDOT films at 2000 rpm. After heat treatment for another 15 min at 60 °C, all the substrates were transferred into the deposition system. Under the pressure of 10$^{-5}$ Torr, the remaining evaporation work of the device was completed. First, DPEPO and TmPyPB were thermally evaporated at a rate of 1.0 and 2.0 Å s$^{-1}$, respectively. Then, the electron-injecting layer LiF was slowly deposited on the organic surface at a rate of 0.15 Å s$^{-1}$. Last, the

aluminum electrode was thermally evaporated at a rate of 1.0–3.0 Å s$^{-1}$. The electroluminescence spectra and luminance of the OLED devices were characterized by a PR670 spectrometer with a Keithley 2400 source meter at room temperature.

A detailed description of the characterization method, synthetic procedures, and simulation information are provided in Supplementary Information, available in the online version of the paper.

## Data availability

The X-ray crystallographic coordinates for the T-TNP structure reported in this study have been deposited at the Cambridge Crystallographic Data Centre (CCDC), under deposition number 2179145. These data can be obtained free of charge from The Cambridge Crystallographic Data Centre via www.ccdc.cam.ac.uk/data_request/cif. Most data generated or analyzed in this article are included in this published article or the Supplementary Information.

Optimized geometries and corresponding energy levels of all emitters are provided in the source data file. Source data are provided with this paper.

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

## Acknowledgements

The financial support of the National Natural Science Foundation of China (Nos. 51922021 and 52273164 (Z.R.) and 52103220 (Y.L.)) and Shandong Provincial Natural Science Foundation (ZR2019ZD50 (S.Y.) and ZR2022ZD37 (Z.R. and Y.L.)) are gratefully acknowledged. We thank Prof. S. Gong (Wuhan University) for his assistance with the angle-dependent /p-polarized emission, as well as associate Prof. D. Yang (South China University of Technology) and Prof. J. Tang (Soochow University) for their assistance with the optical simulation of the devices.

## Author contributions

Z.R., S.Y., & L.Z. initiated and supervised the project. Z.R. and L.H. designed the experiments and L.H. synthesized and characterized the TADF emitters. B.L. provided help in synthesis and performed the single

crystal characterization. L.H. performed photophysical and electrochemical measurements of the TADF emitters. L.H. & Y.L. completed the preparation and characterization of OLED devices. L.H. and Z.Z. performed the computational calculation. Z.R. and L.H. wrote the manuscript and S.Y. revised the manuscript. All authors discussed the progress of the research and reviewed the manuscript.

## Competing interests

The authors declare no competing interests.
