## [Peer Review File · Nature Communications]

Constructing High-Efficiency Orange-Red Thermally Activated Delayed Fluorescence Emitters by Three-Dimension Molecular EngineeringReviewers' Comments:

Reviewer #1:

Remarks to the Author:

The authors developed a series of orange-red TADF emitters for solution-processed OLEDs by introducing different numbers of TADF units on the novel linker trinaphthopropellane (TNP), and discussed the effects of different functional units on photophysical properties and device performances. A solution-processed OLED based on tBu-S-mCP obtained orange-red EL peak emission at 594 nm and a high maximum external quantum efficiency (EQEmax) of 24.74%, which is among the highest levels in the scientific literature. However, in my view, some discussions to realize high-efficiency in solution-processed device are not rational. At this stage, the reviewer has a feeling that this paper is not suitable to be published in Nature Communications. There are several issues described below.

- 1) The authors described: "The introduction of propeller-like shape TNP can indeed separate TADF units and thus ensure the reduced quenching effect resting on the crystal packing pattern". The NAI-based TADF unit comes from previous research by Yang's team (Adv. Mater. 2018, 30, 1704961; Adv. Mater. 2019, 1901404). Compared to the previous research, the PLQY of the emitters in this paper does not have improvement, so can this design really suppress the concentration quenching? The authors should give further experimental evidences and explanations. To clarify the effect of the novel linker, it is important to add control materials such as monomer NAI for comparison.
- 2) Figure 1 is not easy to understand. The chemical structures of the materials are too small. The authors should modify the picture to make the design strategy and molecular structure clear.
- 3) The authors discuss the packing mode of the molecule based on the single crystal structure of the molecule and believe that exciton quenching is effectively suppressed. However, amorphous films are used in both physical property testing and OLED devices, and the emitters are doped in the host material. Therefore, it is no rational to simply use the single-crystal structure to discuss the molecular packing mode of the emitters. To demonstrate suppression of exciton quenching, the authors should provide and discuss the PLQY of the emitters at different doping concentrations.
- 4) In the physical property and OLED device, the host materials of the doped films are completely different: mCPCN is used in the PLQY; PMMA is used in the transient PL and mCPCN/PVK is used in the device, which makes it difficult to understand and analyze the data. The authors should use one host material consistently.
- 5) It is necessary to discuss the relationship between PLQY and EQE for different emitters, but the authors only listed the result in the main text without a careful discussion. For example, the PLQY of a single TADF series emitter is not high (41-58%), but it shows a very high EQE (16.7- 24.7%), why? The authors should provide rational reasons for the obtained EQE. In addition, the molecular orientation of the emitters can boost the light out-coupling efficiency. These emitters showed horizontal molecular orientation?
- 6) The authors should include the values of V, CE, PE, and EQE at 100 cdm⁻² and 1000 cdm⁻² in Table2, only the maximum values are no meaning.
- 7) Thermophysical data (glass transition temperature, decomposition temperature) for all emitters should be provided.
- 8) The elemental analyses or HPLC purities should be shown for the final product.
- 9) Why the authors used mCPCN as host material? For the red emitter, energy-gap of mCPCN is too wide for red emission.

10) The obtained L-V characteristics shown in Figure 6(c) are strange. The authors should provide clearer L-V characteristics and J-V characteristics and comment on them.

Reviewer #2:

Remarks to the Author:

In this study, the authors proposed an interesting three-dimension molecular engineering strategy to significantly enhance the efficiency of orange-red thermally activated delayed fluorescence emitters with well-modulated excited state natures. Detailed photophysical investigations and DFT calculations of the ground and excited states provide a powerful support for the proposed mechanism. With this strategy, a high-efficiency orange-red emitter peaked at 604 nm with high photoluminescence quantum yield of 58.22% can be obtained. Solution-processable organic light-emitting diodes exhibit a maximum external quantum efficiency of 24.74%, which is in the first tier so far. Additionally, this work has expanded the application scope of designing the other high-performance TADF emitters. I believe it can attract a broad readership in TADF materials and organic optoelectronic fields. Therefore, I recommend this work to be published on Nature Communications after addressing the following concerns.

1. Figure 1 should be slightly modified. With host modification, tBu-modified TADF units should also be added into Single-TADF and Double-TADF series.

2. The authors stated that the obtained raw material tribromotrinaphtho [3,3,3] propellane (TNP-3Br) is a mixture of isomers. I suggest the authors to give the detail analysis, such as ¹H NMR result.

3. In photophysical properties section, the assignments of UV absorption peaks should cite the references for clarity.

4. Actually, the onset positions of the fluorescence at 77K and phosphorescence spectra were used to calculate ΔE_{ST} . Please correct it in the text.

5. The significant digitals in table 1 should be unified. It is too much for PLQY, ϕ_p and ϕ_d .

6. The authors should explain "the DF ratio of tBu-S-mCP increases from 61% at 77 K to 82.2% at 300 K in Figure 4(f)." in detail.

Reviewer #3:

Remarks to the Author:

Developing high-efficiency solution-processable orange-red thermally activated delayed fluorescence (TADF) emitters remains to be a challenge. In this manuscript, the authors proposed a new strategy for the design of orange-red TADF molecules from the perspective of molecular packing and the regulation of excited states. By introducing the trinaphtho[3,3,3]propellane unit into the TADF units, a series of orange-red emitting TADF emitters were synthesized. With this strategy, the performance of orange-red light TADF molecules was improved. Among them, OLEDs based on tBu-S-mCP achieved an efficiency of 24%. The work is quite innovative in this field, which will guide the design of other high-performance TADF emitters. Therefore, I recommend its publication after addressing the following concerns.

1. In part of Introduction, the author mentioned "EQEmax values have achieved over 30% for red OLEDs ..." What is the difference of molecular design for solution-processed versus vacuum-deposited materials? Why do solution-processed devices perform the inferior efficiency comparing to vacuum-deposited ones?

2. As a similar three-dimensional structure, there have been some studies combining triptycene units with TADF units. Do TNP units have some unique advantages over triptycene groups? Some discussions may be added.

3. The authors believe that the molecular design strategy incorporating TNP units is applicable to orange-red emitting molecules. Can this design strategy be applied to other light-emitting TADF molecules, such as blue or green emitters?
4. The authors describe the regulation of excited states in detail from the perspective of natural transition orbits. However, the Marcus-Hush theory cited herein is not complete enough, and the modified Marcus-Hush theory should be used.
5. There are too many significant figures for data of PLQY, ϕ_p and ϕ_d in Table 1.
6. In the OLED device section, the data of power efficiency are missing. The authors are suggested to provide the data in the manuscript or supporting information.

Reviewer #1

Comments: The authors developed a series of orange-red TADF emitters for solution-processed OLEDs by introducing different numbers of TADF units on the novel linker trinaphthopropellane (TNP), and discussed the effects of different functional units on photophysical properties and device performances. A solution-processed OLED based on tBu-S-mCP obtained orange-red EL peak emission at 594 nm and a high maximum external quantum efficiency (EQE_{\max}) of 24.74%, which is among the highest levels in the scientific literature. However, in my view, some discussions to realize high-efficiency in solution-processed device are not rational. At this stage, the reviewer has a feeling that this paper is not suitable to be published in Nature Communications. There are several issues described below.

Response: Many thanks for positive comments.

1. The authors described: "The introduction of propeller-like shape TNP can indeed separate TADF units and thus ensure the reduced quenching effect resting on the crystal packing pattern". The NAI-based TADF unit comes from previous research by Yang's team (*Adv. Mater.* 2018, 30, 1704961; *Adv. Mater.* 2019, 1901404). Compared to the previous research, the PLQY of the emitters in this paper does not have improvement, so can this design really suppress the concentration quenching? The authors should give further experimental evidences and explanations. To clarify the effect of the novel linker, it is important to add control materials such as monomer NAI for comparison.

Response: Many thanks for questions. In the previous research you mentioned (*Adv. Mater.* 2018, 30, 1704961), the PLQY of NAI-based TADF emitters were measured to be 59.9% in nitrogen. But we previously determined PLQY in air atmosphere, in which triplet excitons were quenched by oxygen to result in a relatively low PLQY. Therefore, the PLQY of our emitters does not have improvement compared with the reference. To further equitably compare the PLQY of our emitters with the reported one, we integrated the steady state emissions both in air and vacuum and calculated and compared the PLQYs in vacuum ($PLQY_{vac}$) of our emitters and the control emitter NAI according to the following equation:

$$PLQY_{vac} = \frac{I_{vac}}{I_{air}} PLQY_{air}$$

where, I_{vac} and I_{air} is the integrate of the steady state emission in vacuum and air atmospheres, respectively.

Firstly, we performed the PLQY of all emitters (2%) doped into the unified hosts (93% mCP-CN and 5% PVK) by this method. As shown in Figure R1, the PLQY of NAI is calculated to be 60.3%, which is nearly consistent with the reference (59.9% in Nitrogen atmosphere, *Adv. Mater.* 2018, 30, 1704961). Under the exactly same conditions, tBu-S-mCP displays the significantly high PLQY (70.9%). The PLQYs of the other emitters have been updated as shown in Table 1 of the revised manuscript. This result fully demonstrates TNP unit can effectively suppress the concentration quenching of excitons.

Figure R1. The steady-state spectra in vacuum or air atmospheres and PLQY in vacuum of (a) tBu-S-mCP, and (b) NAI.

2. Figure 1 is not easy to understand. The chemical structures of the materials are too small. The authors should modify the picture to make the design strategy and molecular structure clear.

Response: Thanks for the suggestions. The details of Figure 1 have been modified as shown the following.

Figure 1. Chemical structure of TNP and molecular engineering pathway. Three reactive bromo-sites on the TNP molecule enable the introduction of the TADF unit (NAI-DMAC) or host units (mCP or DMAC) onto it upon request. Three series of molecules were successfully prepared, named as single-TADF series (S-mCP, S-DMAC, tBu-S-mCP, tBu-S-DMAC), double-TADF series (D-mCP, D-DMAC, tBu-D-mCP) and triple-TADF series (T-TNP, tBu-T-TNP) according to the number and type of modified TADF and host units.

3. The authors discuss the packing mode of the molecule based on the single crystal structure of the molecule and believe that exciton quenching is effectively suppressed. However, amorphous films are used in both physical property testing and OLED devices, and the emitters are doped in the host material. Therefore, it is no rational to simply use the single-crystal structure to discuss the molecular packing mode of the emitters. To demonstrate suppression of exciton quenching, the authors should provide and discuss the PLQY of the emitters at different doping concentrations.

Response: Thanks for the suggestions. The PLQYs of tBu-S-mCP with the different doping concentrations have been characterized. Similarly, PLQY in vacuum can be converted by PLQY in air and the ratio of steady-state spectral integration in vacuum to air. Taking tBu-S-mCP as an example, 2 wt%, 5 wt% and 10 wt% tBu-S-mCP are doped in 5 wt% PVK with corresponding concentrations of mCP-CN. The corresponding results

are shown in the Supplementary Figure 5. When the doping concentration increases from 2 wt% to 10 wt%, the PLQY of tBu-S-mCP in vacuum decreases by about 7% (from 70.9% to 63.1%). However, the PLQY of the control emitter (NAI) decreases by about 14% with a doping concentration of from 1.5 wt% to 6 wt% (*Adv. Mater.* **30**, 8 (2018)). Compared to the control emitter NAI, the PLQY of tBu-S-mCP is significantly less affected by the doping concentration. In combination with the characterization of single crystal, it is well demonstrated that the architecture of TNP can effectively suppress exciton quenching. This is an effective design strategy to enhance the performance of orange-red TADF emitters. The corresponding discussion has been added into the photophysical properties section of the revised manuscript.

Supplementary Figure 5. The PLQYs of tBu-S-mCP with the different doping concentrations. (a) The PLQYs of tBu-S-mCP at 2 wt%, 5 wt% and 10 wt% tBu-S-mCP doped in 5 wt% PVK and corresponding concentrations of mCP-CN in vacuum and air atmospheres. The steady-state spectra in vacuum and air atmospheres of (b) 5 wt% tBu-S-mCP doped films and (c) 10 wt% tBu-S-mCP doped films.

4. In the physical property and OLED device, the host materials of the doped films are completely different: mCPCN is used in the PLQY; PMMA is used in the transient PL and mCPCN/PVK is used in the device, which makes it difficult to understand and analyze the data. The authors should use one host material consistently.

Response: Thanks for the suggestions. We united the host mCP-CN/PVK for key photophysical characterizations and OLEDs measurement, including PLQY and transient PL. The relevant photophysical results were re-analyzed based on the doped films (2 wt% emitters doped in 93 wt% mCP-CN and 5 wt% PVK). Although the host material and concentration have been changed, the relationship between the molecular structure and its properties has not been changed, so the trend of data remains unchanged. The relevant photophysical characterization is also reanalyzed as shown the Table 1 of the revised manuscript.

Table 1. Summary of photophysical properties of the nine emitters.

	PLQY /% ^(a)	τ_p /ns ^(b)	τ_d / μ s ^(c)	Φ_p /% ^(d)	Φ_d /% ^(e)	k_p / 10^6 s ^{-1(f)}	k_{DF} / 10^5 s ^{-1(g)}	k_{ISC} / 10^6 s ^{-1(h)}	k_{RISC} / 10^5 s ⁻¹⁽ⁱ⁾	k_{nr}^s / 10^5 s ^{-1(j)}
tBu-S-mCP	70.9	23.9	6.41	18.5	81.5	5.48	9.02	4.47	4.88	2.95
tBu-S-DMAc	57.8	23.4	6.65	17.1	82.9	4.22	7.20	3.50	4.21	3.06
S-mCP	56.3	23.4	6.47	19.1	80.9	4.59	7.05	3.72	3.70	3.82
S-DMAc	49.5	24.7	6.49	17.3	82.7	3.47	6.30	2.87	3.64	3.04
tBu-D-mCP	51.7	23.3	5.59	19.2	80.8	4.26	7.47	3.45	3.89	3.96
D-mCP	44.4	25.6	5.79	19.4	80.6	3.35	6.18	2.70	3.19	3.61
D-DMAc	43.9	26.3	5.57	19.5	80.5	3.24	6.35	2.61	3.26	3.54
tBu-T-TNP	43.4	25.9	5.33	20.1	79.9	3.36	6.51	2.69	3.24	3.82

(a) The photoluminescence quantum yields of blended films measured in air condition; (b) The lifetime of prompt fluorescence component; (c) The lifetime of delayed fluorescence component; The ratio of (d) prompt fluorescence (PF) component and (e) delayed fluorescence component determined by fitting transition decay curves; The (f) prompt and (g) delayed fluorescence decay rate, respectively; (h) The rate constants of intersystem crossing; (i) The rate constants of reverse intersystem crossing; (j) The nonradiative rate constants of the S_1 states.

5. It is necessary to discuss the relationship between PLQY and EQE for different emitters, but the authors only listed the result in the main text without a careful discussion. For example, the PLQY of a single TADF series emitter is not high (41-58%), but it shows a very high EQE (16.7- 24.7%), why? The authors should provide rational reasons for the obtained EQE. In addition, the molecular orientation of the emitters can boost the light out-coupling efficiency. These emitters showed horizontal molecular orientation?

Response: Thanks for the suggestions. As shown in Supplementary Figure 11, the emitting dipole orientations of the typical emitters, tBu-S-mCP and tBu-S-DMAc, were determined by angle- and polarization-resolved PL spectroscopy measurements. The both horizontal dipole ratios ($\Theta//$) are fitted to be 73% (fully horizontal dipoles: $\Theta//=100\%$ and isotropic dipole orientation: $\Theta//=67\%$), and the corresponding calculated outcoupling efficiency (η_{out}) is estimated to be around 35.3%, suggesting the horizontal molecular orientation. Combined with their PLQY in vacuum (70.9% for tBu-S-mCP and 57.8% for tBu-S-DMAc), the corresponding theoretical maximum external quantum efficiencies (EQE_{cal}) are calculated to be 25.0% for tBu-S-mCP and 20.4% for tBu-S-DMAc, respectively ($EQE_{cal} = PLQY_{vac} \times \eta_{out}$). Therefore, the obtained EQE_{max} is rational for tBu-S-mCP (24.7%) and tBu-S-DMAc (16.9%). The corresponding discussion has also been added into the revised manuscript.

Supplementary Figure 11. Molecular orientation characterization. Angle- and polarization-resolved PL spectra of (a) tBu-S-mCP and (b) tBu-S-DMAc (2 wt% doped in 5 wt% PVK and 93 wt% mCP-CN) films. Measured (symbols) and simulated curves (lines and dotted lines) with horizontal dipole ratios ($\Theta//$) are marked.

6. The authors should include the values of V, CE, PE, and EQE at 100 $cd\ m^{-2}$ and 1000 $cd\ m^{-2}$ in Table 2, only the maximum values are no meaning.

Response: Thanks for the suggestions. The corresponding data for Voltage, CE, PE, and EQE values at 100 $cd\ m^{-2}$ and 1000 $cd\ m^{-2}$ have been added to Table 2 of the revised manuscript, as follows.

Table 2. EL properties of solution-processed OLEDs based on orange-red emitters.

	EL_{peak} (nm) ^(a)	$V_{on, 100, 1000}$ (V) ^(b)	L_{max} ($cd\ m^{-2}$) (c)	$CE_{max, 100, 1000}$ ($cd\ A^{-1}$) ^(d)	$PE_{max, 100, 1000}$ ($lm\ W^{-1}$) ^(e)	$EQE_{max, 100, 1000}$ (%) ^(f)	CIE [x,y] ^(g)
tBu-S-mCP	594	5.4, 5.8, 6.8	1348	59.3, 34.2, 6.57	33.3, 18.5, 3.13	24.74, 14.89, 3.02	0.54,0.44

tBu-S-DMAc	596	5.2, 5.6, 6.4	2047	38.5, 38.5, 6.93	21.6, 21.6, 3.40	16.69, 16.69, 3.22	0.54,0.43
S-mCP	594	5.4, 5.8, 6.6	1266	45.3, 22.2, 5.36	26.3, 12.1, 2.55	19.65, 9.67, 2.53	0.52,0.43
S-DMAc	598	5.2, 6.0, 6.6	2175	39.6, 39.6, 9.62	20.7, 20.7, 4.58	17.61, 17.61, 4.51	0.54,0.43
tBu-D-mCP	604	5.4, 5.8, 6.8	1349	33.5, 27.8, 5.40	18.8, 15.0, 2.49	16.35, 14.01, 2.90	0.56,0.41
D-mCP	604	5.4, 5.8, 6.2	1710	28.0, 23.4, 6.79	15.7, 12.7, 3.44	14.17, 12.17, 3.68	0.56,0.41
D-DMAc	604	5.2, 5.8, 6.6	1881	26.2, 20.4, 6.71	14.7, 11.1, 3.20	13.22, 10.56, 3.69	0.56,0.41
tBu-T-TNP	608	5.2, 5.8, 7.0	1305	25.9, 8.21, 3.82	15.1, 4.45, 1.71	14.42, 4.71, 2.31	0.57,0.40
T-TNP	608	5.2, 5.6, 6.6	1376	26.3, 18.4, 4.68	15.3, 10.3, 2.23	14.93, 10.7, 2.89	0.57,0.39

(a) The peak value of electroluminescence at maximum brightness; (b) Turn-on voltage at 1 cd m⁻² and voltages at 100 and 1000 cd m⁻²; (c) Maximum luminance; (d) Maximum current efficiency and values at 100 and 1000 cd m⁻² (e) Maximum power efficiency and values at 100 and 1000 cd m⁻²; (f) Maximum external quantum efficiency and values at 100 and 1000 cd m⁻²; (g) Coordinates of Commission Internationale de L'Eclairage.

7. Thermophysical data (glass transition temperature, decomposition temperature) for all emitters should be provided.

Response: Thanks for the suggestions. Thermophysical data (glass transition temperature, decomposition temperature) for all emitters have been provided as shown in Supplementary Figure 12. According to the DSC characterization, all emitters show no obvious glass transition, melting or crystallization behavior in the range from room temperature to 300 °C. Their initial thermal decomposition temperatures are all above 400°C, which indicates that all emitters have good thermal stability and meet the requirement for the device preparation process.

Supplementary Figure 12. Thermal properties characterization. Thermal gravimetric analysis of (a) Single-TADF series, (b) Double-TADF series, (c) Triple-TADF series. The insets are the differential scanning calorimetry characterizations of the emitters.

8. The elemental analyses or HPLC purities should be shown for the final product.

Response: Thanks for the suggestions. We performed HPLC measurements to confirm the purities of our emitters as shown in Supplementary Figure 51. All final products display a single peak with an integral area of over 99.9%, indicating their high purities.

Supplementary Figure 51. High performance liquid chromatography of nine emitters.

9. Why the authors used mCPCN as host material? For the red emitter, energy-gap of mCPCN is too wide for red emission.

Response: Thanks for comment. The energy gap of mCP-CN is 3.56 eV with a high triplet state energy level of 3.03 eV (*J. Mater. Chem.* 22, 16114–16120 (2012)). Therefore mCP-CN is a suitable host material for almost all visible light emitting materials. The UV-Vis absorption of the TNP series emitters decreases significantly above 360 nm and is extremely weak in the charge transfer absorption above 400 nm. Using a wide bandgap mCP-CN host ensures adequate energy transfer from host to the emitters, especially at a low doping ratio of emitter. At the same time, the cyano group in mCP-CN can effectively balance hole-electron injection and transport properties. In addition, intermolecular electrostatic dipole-dipole interactions between mCP-CN host and guest molecules have been reported to improve device performance (*Adv. Funct. Mater.* 32, 2203022(2022)). Therefore, mCP-CN was used as the host material for our emitters. A description about mCP-CN has been added into the device section of the revised manuscript.

10. The obtained L-V characteristics shown in Figure 6(c) are strange. The authors should provide clearer L-V characteristics and J-V characteristics and comment on them.

Response: Thanks for the suggestions. The clearer L-V characteristics and J-V characteristics are provided as follow. In the curves of current density–voltage–luminance, the turn-on voltages of all devices are similar, ranging from 5.2 to 5.4 V. At around 7 V, all devices achieve the stable maximum brightness over 1000 cd m⁻². The current density maintains a uniform increase with increasing voltage. The figures have been enlarged in the revised manuscript.

Figure R2. Curves of current density–voltage–luminance of OLED device. (a) Single-TADF series; (b) Double-TADF series; (c) Triple-TADF series

Reviewer #2

Comments: In this study, the authors proposed an interesting three-dimension molecular engineering strategy to significantly enhance the efficiency of orange-red thermally activated delayed fluorescence emitters with well-modulated excited state natures. Detailed photophysical investigations and DFT calculations of the ground and excited states provide a powerful support for the proposed mechanism. With this strategy, a high-efficiency orange-red emitter peaked at 604 nm with high photoluminescence quantum yield of 58.22% can be obtained. Solution-processable organic light-emitting diodes exhibit a maximum external quantum efficiency of 24.74%, which is in the first tier so far. Additionally, this work has expanded the application scope of designing the other high-performance TADF emitters. I believe it can attract a broad readership in TADF materials and organic optoelectronic fields. Therefore, I recommend this work to be published on Nature Communications after addressing the following concerns.

Response: Many thanks for positive comments.

1. Figure 1 should be slightly modified. With host modification, tBu-modified TADF units should also be added into Single-TADF and Double-TADF series.

Response: Thanks for good suggestions. The details of Figure 1 have been modified as advised. (see answer of reviewer 1 above)

2. The authors stated that the obtained raw material tribromotrinitro [3,3,3] propellane (TNP-3Br) is a mixture of isomers. I suggest the authors to give the detail analysis, such as ¹H NMR result.

Response: Thanks for advice. The ¹H NMR result of the obtained raw material tribromotrinitro [3,3,3] propellane has been add in Supplementary Figure 14. From the results of the ¹H NMR, the three hydrogen atoms with chemical shifts of 8.01 - 8.08 ppm show irregular splitting. This is due to the uncertainty in the position of the end group bromine, so that the three blades of TNP are not completely symmetrical. The ¹H NMR of this raw material is consistent with the description of TNP-3Br in the literature (*ChemPlusChem*, 82, 1006-1009 (2017)).

Supplementary Figure 14. ¹H NMR spectrum of TNP-3Br in chloroform-*d*.

3. In photophysical properties section, the assignments of UV absorption peaks should cite the references for clarity.

Response: Thanks for good suggestion. Corresponding references have been added to the revised manuscript. (Ref. 29 *Adv. Mater.* 30, 8 (2018) & Ref. 46 *Adv. Mater.* 26, 4050-4055 (2014))

4. Actually, the onset positions of the fluorescence at 77 K and phosphorescence spectra were used to calculate ΔE_{ST} . Please correct it in the text.

Response: Thanks for comment. The calculation of ΔE_{ST} has been corrected in the photophysical properties section of the revised manuscript. “The ΔE_{ST} s of all emitters were measured by fluorescence and phosphorescence spectra in toluene solution at 77 K. The onset positions of the fluorescence and phosphorescence spectra at 77 K were used as the singlet and triplet energy levels, respectively.”

5. The significant digits in table 1 should be unified. It is too much for PLQY, ϕ_p and ϕ_d .

Response: Thanks for good suggestions. Three significant digits were kept for PLQY, ϕ_p and ϕ_d .

6. The authors should explain “the DF ratio of tBu-S-mCP increases from 61% at 77 K to 82.2% at 300 K in Figure 4(f).” in detail.

Response: Thanks for question. The DF ratio of tBu-S-mCP increases from 61% at 77 K to 82.2% at 300 K, which indicates the much more triplet exciton of tBu-S-mCP contributes to the luminescence at 300 K than that of 77 K. That is, tBu-S-mCP presents a more efficient RISC process at 300 K than 77 K, which is consistent with the nature of thermal activation. The related discussion has been added into the revised manuscript.

Reviewer #3

Comments: Developing high-efficiency solution-processable orange-red thermally activated delayed

fluorescence (TADF) emitters remains to be a challenge. In this manuscript, the authors proposed a new strategy for the design of orange-red TADF molecules from the perspective of molecular packing and the regulation of excited states. By introducing the trinaphtho[3,3,3]propellane unit into the TADF units, a series of orange-red emitting TADF emitters were synthesized. With this strategy, the performance of orange-red light TADF molecules was improved. Among them, OLEDs based on tBu-S-mCP achieved an efficiency of 24%. The work is quite innovative in this field, which will guide the design of other high-performance TADF emitters. Therefore, I recommend its publication after addressing the following concerns.

Response: Many thanks for positive comments.

1. In part of Introduction, the author mentioned “EQE_{max} values have achieved over 30% for red OLEDs ...” What is the difference of molecular design for solution-processed versus vacuum-deposited materials? Why do solution-processed devices perform the inferior efficiency comparing to vacuum-deposited ones?

Response: Thanks for your questions. From the viewpoint of molecular design, the emitters designed for solution-processed OLEDs must meet good solubility and film-forming properties. Consequently, some solubility-enhanced groups are frequently introduced into emitters, such as alkyl groups. Furthermore, the rigid architecture of emitters should also be reduced since it can cause strong crystallization or significant aggregation during solution processing, which is detrimental to the stability of OLEDs (*Adv. Sci.* 8, 2101326 (2021); *Nat. Rev. Mater.* 3, 18020 (2018)). In addition, the much more balanced carrier transport properties are required for solution-processed emitters than vacuum-deposited ones due to the disorder of solution-processed films (*J. Mater. Chem. C* 7, 12321-12327 (2019)). Also, good compatibility with the host will facilitate the improvement of OLEDs performance for solution-processed emitters (*Adv. Mater.* 34, 2110547 (2022)).

The performance gap between solution-processed and vacuum-deposited OLEDs for orange-red emitters is mainly caused by the different microstructure of the light-emitting layer. Generally, solution-processed films present the much more defects, the higher disorder and thus the lower carrier transport properties than vacuum-deposited films (*Adv. Sci.* 8, 2101326 (2021)). Therefore, the performance of solution-processed OLEDs is still significantly worse than that of vacuum-deposited ones.

2. As a similar three-dimensional structure, there have been some studies combining triptycene units with TADF units. Do TNP units have some unique advantages over triptycene groups? Some discussions may be added.

Response: Thanks for comments. TNP unit possesses the larger blades, which is more planar in the single blade direction than the triptycene. This characteristic endows TNP units the ability to preventing stacking, which is desirable for TADF emitters (*Chem. Sci.* 10, 4951-4958 (2019)). Our experimental single-crystal data for T-TNP also suggest that the TADF units can be kept in the cavities between two adjacent molecules, illustrating the specificity of TNP as an intermediate linking unit. It is different from the previously reported triptycene units.

3. The authors believe that the molecular design strategy incorporating TNP units is applicable to orange-red emitting molecules. Can this design strategy be applied to other light-emitting TADF molecules, such as blue or green emitters?

Response: Thanks for good question. The triplet energy level of TNP plays decisive role in whether our design strategy can be applied to blue or green TADF emitters. Generally, the triplet energy level of TNP is required to be higher than TADF units to avoid energy transfer back and thus quenching of emission. Therefore, we calculated the triplet energy level of TNP-3Br by the onset positions of the phosphorescence spectra at 77 K.

As shown in Figure R3, the triplet state energy level of TNP-3Br is evaluated to be 2.50 eV, which is clearly too low for blue and green emitters. However, for yellow and orange-red emitting materials, the triplet state energy level of TNP is high enough, and thus this molecular design strategy is universal for orange-red emitters.

Figure R3. Phosphorescence spectra of TNP-3Br detected in toluene at 77 K.

4. The authors describe the regulation of excited states in detail from the perspective of natural transition orbits. However, the Marcus-Hush theory cited herein is not complete enough, and the modified Marcus-Hush theory should be used.

Response: Thanks for the suggestions. To fully represent the excited state properties, especially the fine tuning of the ³LE states, the second order vibronic coupling mechanism is involved and discussed in the revised manuscript. According to the second order vibronic coupling mechanism, the ³LE state can act as a mediator role which can assist the flipping of the exciton from triplet to the singlet (*Nat. Commun.* 7, 13680 (2016)). In our case, comparing with the other emitters, tBu-S-mCP exhibits the tiny energy difference of both ¹CT-³CT and ¹CT-³LE due to the fine modulation by the tert-butyl and mCP host groups. According to the second order vibronic coupling mechanism, the smaller energy difference is conducive to the promotion of efficient RISC processes, so tBu-S-mCP stands out among these emitters with the best performance. The corresponding discussions have been added in the revised manuscript.

5. There are too many significant figures for data of PLQY, ϕ_p and ϕ_d in Table 1.

Response: Thanks for good suggestions. Three significant digits were kept for PLQY, ϕ_p and ϕ_d .

6. In the OLED device section, the data of power efficiency are missing. The authors are suggested to provide the data in the manuscript or supporting information.

Response: Thanks for the suggestions. The corresponding current efficiency and power efficiency have been added into Table 2 of the revised manuscript.

We thank you and our reviewers for taking time to handle and review our manuscript.

Zhongjie Ren

Reviewers' Comments:

Reviewer #1:

Remarks to the Author:

Although the authors modified the article according to the reviewer comments, some key points, such as the analysis of the photophysical properties of emitters and how to achieve high EL efficiencies, are not clearly clarified. In addition, some data and methods are not rational. The reviewer has a feeling that the story and innovation of this paper are not suitable to be published in Nature Communications. There are several issues described below.

1. The authors re-measured PLQY in vacuum (PLQY_{vac}) and compared it with PLQY in air (PLQY_{air}). In air atmosphere, triplet excitons are quenched by oxygen, so I_{air}/I_{vac} corresponds to the prompt fluorescence component (j_p). However, in Figure R1, I_{air}/I_{vac} is 0.66, while in Table 1, the j_p of tBu-S-mCP is only 0.185, which are contradictory data.

2. The authors said "the corresponding calculated outcoupling efficiency (η_{out}) is estimated to be around 35.3%, suggesting the horizontal molecular orientation." Does this calculated outcoupling efficiency correspond to the outcoupling efficiency of the device? In order to obtain the outcoupling efficiency of the device, the authors should perform simulation calculations on the device structure used and give the calculation method.

3. In the supporting information, about the device fabrication method, the authors first spin-coated PEDOT:PSS and heat-treated at 120 °C to remove residual water. Subsequently, the authors used chlorobenzene solution to prepare emissive film on PEDOT:PSS surface and only heat-treated at 60 °C for 15 minutes. However, the boiling point of chlorobenzene is 132 °C, and the residual chlorobenzene in the film cannot be completely removed by this method. Based on this method, the reviewer believes that residual chlorobenzene remains in all the emissive films, which may seriously affect the reliability of the data, especially for the photophysical properties and the EL properties.

Reviewer #2:

Remarks to the Author:

The authors seem to have correctly addressed all of my concerns. I believe it is OK to be accepted for publication now.

Reviewer #3:

Remarks to the Author:

The revised manuscript is recommended for publication.

Reviewer #1

Comments: Although the authors modified the article according to the reviewer comments, some key point, such as the analysis of the photophysical properties of emitters and how to achieve high EL efficiencies, are not clearly clarified. In addition, some data and methods are not rational. The reviewer has a feeling that the story and innovation of this paper are not suitable to be published in Nature Communications. There are several issues described blow.

Response: Many thanks for your comments. We think we have clarified the relationship between the photophysical properties of emitters and high EL efficiencies. The obtained high EL efficiencies may be assigned to the high PLQY and the relatively high light out-coupling efficiency caused by a higher molecular dipole orientation. Furthermore, the high PLQY is attributed to both the decreased annihilation of long-lived triplet excitons benefiting from the unique hexagonal stacking architecture of TNPs and the enhanced spin-orbit coupling by regulating the excited state nature. In addition, the rationality of some data and methods you concerned are addressed in detail as shown in the following.

1. The authors re-measured PLQY in vacuum ($PLQY_{vac}$) and compared it with PLQY in air ($PLQY_{air}$). In air atmosphere, triplet excitons are quenched by oxygen, so I_{air}/I_{vac} corresponds to the prompt fluorescence component (jp). However, in Figure R1, I_{air}/I_{vac} is 0.66, while in Table 1, the jp of tBu-S-mCP is only 0.185, which are contradictory data.

Response: Thanks for question. Firstly, in air atmosphere, if all triplet excitons are quenched by oxygen, the I_{air}/I_{vac} obtained from steady-state PL emission corresponds to the prompt fluorescence component. However, triplet excitons usually cannot be quenched completely in air, which is possibly related to the permeability of oxygen into the films (*Chem. Sci.* 2018, 9, 6150). Therefore, in these cases, I_{air}/I_{vac} obtained from steady-state PL emission containing a certain amount of delayed fluorescence component (shown in Figure R1) cannot simply equal to the fitted prompt fluorescence component from transient decay curves in vacuum. That is, I_{air}/I_{vac} obtained from steady-state PL emission should be higher than the fitted prompt fluorescence component from transient decay curves in air. In our case, the doped film of tBu-S-mCP still retains both transient and delayed components from transient decay curves in air (Figure R1), indicating triplet excitons still contribute the emission in air (*Chem. Eng. J.* 2022, 435, 134924). Therefore, I_{air}/I_{vac} (0.66) obtained from steady-state PL emission should be higher than the fitted prompt fluorescence component (0.185) from transient decay curves in vacuum, and these two data are not contradictory. Therefore, we calculated the PLQY in vacuum is reasonable by using PLQY in air and the I_{air}/I_{vac} obtained from steady-state PL emission.

Figure R1. Fluorescence decay of the tBu-S-mCP doped films detected in air.

2. The authors said “the corresponding calculated outcoupling efficiency (η_{out}) is estimated to be around 35.3%, suggesting the horizontal molecular orientation.” Does this calculated outcoupling efficiency correspond to the outcoupling efficiency of the device? In order to obtain the outcoupling efficiency of the device, the authors should perform simulation calculations on the device structure used and give the calculation method.

Response: Thanks for suggestion. The optical simulation was performed to predict the maximum EQE of tBu-S-mCP as functions of PLQY of the emitter under the assumption of no electrical loss. The device structure shown in Figure R2b was used for the simulation under the assumption of perfect electrical balance. The recombination zone was assumed to be located in the middle of the emission layer. The input parameters include refractive index value, extinction coefficient, thickness of each layer values, as well as photoluminescence spectrum of the emitting layer. The refractive indexes (Figure R2a) and extinction coefficient of tBu-S-mCP doped film was measured by variable angle spectroscopic ellipsometry (VASE). The other refractive index values were obtained in the literature (*J. Mater. Chem. A* 2013, 1, 1770; *ACS Nano* 2015, 9, 7553). The calculated maximum EQE according to the PLQY for device without considering molecular orientation as shown in Figure R2c.

Furthermore, considering that the ratio of horizontal dipole orientation (Θ) of the light-emitting layer is 73%, the external coupling output is favorable. The increased ratio of Θ 73% versus 67% was calculated using finite difference time domain (FDTD) approach by Lumerical FDTD Solutions 8.7.3. An emission dipole with a horizontal orientation (parallel to the substrate) was placed at middle positions. All photons emitted from the dipole into the glass substrate was integrated in the half-space bounded by the reflecting Al cathode. To compare the relative outcoupling efficiency, Gaussian-oscillating dipole pulse with fixed photon number was set for monitoring the light intensity. The device structure and layer parameters used are shown in Figure R2b. The front view of luminous dipole emits light to the outside of the substrate were shown in Figure R2d. The luminous dipole is dotted, so this light field map is roughly a circle. It can be seen that Θ of 73% is significantly enhanced compared to 67% and the light out-coupling efficiency with Θ of 73% is 1.185 times higher than that of 67% (Figure R2e).

Based on the simulated increased ratio of light output for different wavelengths and the calculated EQE value in Figure R2c, the relationship between EQE and PLQY with emission wavelength can be obtained at a Θ of 73%, as shown in Figure R2f. With PLQY of 70.9% and emission peak at 594 nm, the calculated EQE_{max} is 25.02%. According to equation: EQE_{cal} = PLQY_{vac} × η_{out} , η_{out} of the device based on tBu-S-mCP is 35.3%.

These results have been added into the Supplementary Figure 13.

Figure R2. (a) Refractive indexes of tBu-S-mCP doped films measured by variable angle spectroscopic ellipsometry (VASE) (b) Layer structure of the device with the corresponding refractive indexes. (c) The calculated maximum EQE according to the PLQY for device without considering molecular orientation. (d) The front view of luminous dipole emits light to the outside of the substrate with different ratio of horizontal dipole orientation. (e.) The increased ratio of Θ 73% versus 67% was calculated using finite difference time domain (FDTD) approach by Lumerical FDTD Solutions 8.7.3 at different wavelengths. (f) The calculated maximum EQE according to the PLQY and wavelength for device with ratio of horizontal dipole orientation 73% and the increased ratio of Θ 73% versus 67% (1.185).

3. In the supporting information, about the device fabrication method, the authors first spin-coated PEDOT:PSS and heat-treated at 120 °C to remove residual water. Subsequently, the authors used chlorobenzene solution to prepare emissive film on PEDOT:PSS surface and only heat-treated at 60 °C for 15 minutes. However, the boiling point of chlorobenzene is 132 °C, and the residual chlorobenzene in the film cannot be completely removed by this method. Based on this method, the reviewer believe that residual chlorobenzene remains in all the emissive films, which may seriously affect the reliability of the data, especially for the photophysical properties and the EL properties.

Response: Thanks for question. We chose the heat-treated procedure for spin-coated films from chlorobenzene solution (60 °C for 15 minutes) according to the references. For examples, no annealing for spin-coated films from chlorobenzene solution (*Nat. Commun.* 2022, 13, 3845); annealing at 343 K for spin-coated films from chlorobenzene solution (*Nat. Commun.* 2019, 10, 5307); annealing at 60 °C for spin-coated films from chlorobenzene solution (*Synth. Met.* 2019, 279, 116856; *J. Mater. Chem. C* 2019, 7, 11109). In some cases, low annealing temperatures can guarantee film quality. We think chlorobenzene solvent can be removed completely due to the subsequent vacuum vapor deposition for the emitting layer under ca. 10^{-5} Torr. According to Trouton's rule (*Nature* 1945, 155, 274) and Clausius-Clapeyron equation (*Environ. Sel. Technol.* 1982, 16, 645):

$$\ln \frac{P_2}{P_1} = \frac{\Delta_{vap} H_m^*}{R} \left(\frac{1}{T_1} - \frac{1}{T_2} \right)$$

where P is the vapor pressure of the gas and $\Delta_{vap} H_m^*$ is the heat enthalpy of evaporation of the liquid, R is the ideal gas constant and T is the temperature. The boiling point of chlorobenzene is 405.15 K at 1 atm and

the boiling point of chlorobenzene at 10^{-5} torr (0.0076 atm) is ca. 277.3 K. Therefore, chlorobenzene is sufficiently removed at room temperature (298.15 K) with 10^{-5} torr.

To demonstrate chlorobenzene of our films has completely been removed, we prepared spin-coated films from chlorobenzene solution and then annealed it at 60 °C for 15 min to detect the elements of films by X-ray photoelectron spectroscopy. During the measurement, the films were also placed in a vacuum chamber with a pressure of less than 10^{-5} torr, which is similar with the process of device preparation. As shown in Figure R3, chlorine cannot be detected. Therefore, we believe that there is no residual solvent to affect material performance under our annealing condition.

Figure R3. X-ray photoelectron spectrum of tBu-S-mCP doped film and the binding energy of detected elements were marked.

Furthermore, we prepared a control OLED, in which tBu-S-mCP doped film spin-coated from chlorobenzene solution was firstly annealed at 60 °C for 15 min and then dried at 40 °C in vacuum for another 30 min to ensure complete solvent removal. As shown in Figure R4, the EQE_{max} of the device remains at 24.3%, which is nearly consistent with the data of OLEDs prepared by the previous procedures. Therefore, we think our device data are reliable.

Figure R4. (a) Curves of current density-voltage-luminance of additional annealed OLED device. (b) Curves of EQE value and Power efficiency versus luminance.

We thank you and our reviewers for taking time to handle and review our manuscript.

Zhongjie Ren

Reviewers' Comments:

Reviewer #1:

Remarks to the Author:

The authors carefully answered the reviewers' questions and supplemented the data. Nevertheless, the reviewer is unable to recommend this work to be published in Nature Communications. This work aims to use reported emitters and functional groups to combine and obtain high-efficiency devices. Therefore, the reviewer considers this paper need to be transferred to a more appropriate journal in the field of optical materials. There are some issues described below:

1. This work prepared nine similar emitters based on the same design strategy, but only tBu-S-mCP showed relatively high PLQY and EQEmax. The EQEmax of all other emitters cannot reach 20%. The reviewer feels that the high EQE exhibited by tBu-S-mCP is a special case, and this design strategy is not easily applicable to the development of other high performance TADF materials.

2. The calculation formula of EQE ($EQE_{cal} = PLQY_{vac} \times \eta_{out}$) used by the author is not accurate enough. In fact, this formula should be described as:

$$EQE_{cal} = IQE \times \gamma \times \eta_{out}$$

Where γ is the charge recombination efficiency that is ideally 100% for fully balanced carrier recombination. However, in solution-processed devices, γ actually difficult to reach 100%, which is caused by the large hole injection barrier and exciton quenching between PEDOT:PSS and EML. For this reason, the experimental value of EQEmax is usually significantly lower than the theoretically calculated value. Therefore, the reviewer believes that the EQEmax (24.7%) of tBu-S-mCP-based device may be overestimated, which may be caused by the device fabricating process, such as black spots on the device, etc.

Reviewer #1

Comments: The authors carefully answered the reviewers' questions and supplemented the data. Nevertheless, the reviewer is unable to recommend this work to be published in Nature Communications. This work aims to use reported emitters and functional groups to combine and obtain high-efficiency devices. Therefore, the reviewer considers this paper need to be transferred to a more appropriate journal in the field of optical materials. There are some issues described below:

Response: Many thanks for recognizing our previous revisions. In fact, there have been many reports on emitting materials and high-efficiency OLEDs in *Nat. Commun.* recently, such as, *Nat. Commun.* **13**, 4876 (2022); *Nat. Commun.* **13**, 1215 (2022); *Nat. Commun.* **12**, 6179 (2021); *Nat. Commun.* **12**, 3640 (2021); *Nat. Commun.* **11**, 1765 (2020); *Nat. Commun.* **11**, 1758 (2020). Therefore, we think our report of high-performance TADF emitters and OLEDs is within the scope of *Nat. Commun.*

1. This work prepared nine similar emitters based on the same design strategy, but only tBu-S-mCP showed relatively high PLQY and EQE_{max}. The EQE_{max} of all other emitters cannot reach 20%. The reviewer feels that the high EQE exhibited by tBu-S-mCP is a special case, and this design strategy is not easily applicable to the development of other high performance TADF materials.

Response: Thanks for comment. Your question just reflect why we did so much molecular engineering on these molecules. Indeed, a lot works are performed to conclude an effective strategy. Finally, the best emitter tBu-S-mCP is filtered out among three series of emitters via introducing three-dimensional architecture of TNP and tuning the excited state natures of emitters by optimizing the type and amount of TADF and host units. Therefore, if TNP units are introduced into other TADF molecules and refined excited state regulation is performed, better performance will be theoretically achieved.

This screening process, that is, the discussion on the relationship between molecular structures and properties, is important to guide the future design and synthesis of other efficient TADF emitters. In detail, the scheme of quenching inhibition by three-dimensional architecture together with the scheme of regulating excited states through host engineering can provide simple and convenient ideas for the design of other emitters. These design strategies are universal and can be easily applied to improve the performance of TADF material. At the same time, these design strategies have no special requirements for TADF molecules and do not involve very complex chemical reactions.

2. The calculation formula of EQE ($EQE_{cal} = PLQY_{vac} \times \eta_{out}$) used by the author is not accurate enough. In fact, this formula should be described as: $EQE_{cal} = IQE \times \gamma \times \eta_{out}$, where γ is the charge recombination efficiency that is ideally 100% for fully balanced carrier recombination. However, in solution-processed devices, γ actually difficult to reach 100%, which is caused by the large hole injection barrier and exciton quenching between PEDOT:PSS and EML. For this reason, the experimental value of EQE_{max} is usually significantly lower than the theoretically calculated value. Therefore, the reviewer believes that the EQE_{max} (24.7%) of tBu-S-mCP-based device may be overestimated, which may be caused by the device fabricating process, such as black spots on the device, etc.

Response: Thanks for comment. Firstly, our device data are repeatable. All emitters are fabricated at least twice under the same conditions, and the deviation is less than 3% in the test of 8 or more pixels during the

device fabrication processes. Secondly, our devices are uniform and no black spots on the devices are found for all devices. Therefore, we believe our device data are reliable.

For the γ (charge recombination efficiency) you mentioned, it is indeed difficult to obtain an absolutely ideal carrier balance for solution-processed devices, so **our actual EQE is lower than EQE_{cal} , which is within a reasonable range.**

In general, in an optimized device including solution-processed device, γ can be considered as 1 (*Nat. Photon.* **16**, 803-810 (2022); *J. Mater. Chem. C*, **7**, 10851-10859 (2019); *Nano Energy* **65**, 104057 (2019)). In fact, our devices have been carefully optimized. Firstly, PVK with good hole-transporting properties is used (*Commun. Mater.* **1**, 81 (2020)). And its HOMO energy level is only 0.4 eV difference with PSS:PEDOT, which is comparable to NPD or TAPC commonly used in vacuum-deposited devices (*Nat. Photon.* **16**, 803-810 (2022); *Pure Appl. Chem.* **87**, 627-638 (2015)). Secondly, DPEPO is also introduced as a hole barrier layer on the other side (*Nano Energy* **59**, 560-568 (2019)), which can effectively enhance the recombination of carriers in the light-emitting layer (*Chem. Sci.* **7**, 2870-2882 (2016)) and thus improve the overall device efficiency. Therefore, during the process of theoretical prediction, we consider our device is fully optimized.

In addition, the theoretically predicted EQE_{max} based on device structure also shows its deviation. In the previous reports, some devices even exhibit the actual higher EQE values than the predicted EQE_{cal} (*Adv. Funct. Mater.* **26**, 7560-7571 (2016); *J. Mater. Chem. C* **5**, 1027-1032(2017); *Chem. Mater.* **27**, 6675-6681(2015)). Therefore, the measured EQE values of our devices are lower than the predicted EQE_{cal} , further indicating the efficiency of our devices is within reasonable limit.

We thank you and our reviewer for taking time to handle and review our manuscript.

Zhongjie Ren